# Substrate engagement by the intramembrane metalloprotease SpoIVFB

**Melanie A. Orlando[1,2], Hunter J. T. Pouillon[1,2], Saikat Mandal ⓘ[1,2], Lee Kroos[1] & Benjamin J. Orlando ⓘ[1]✉**

S2P intramembrane metalloproteases regulate diverse signaling pathways across all three domains of life. However, the mechanism by which S2P metalloproteases engage substrates and catalyze peptide hydrolysis within lipid membranes has remained elusive. Here we determine the cryo-EM structure of the S2P family intramembrane metalloprotease SpoIVFB from *Bacillus subtilis* bound to its native substrate Pro-σ[K]. The structure and accompanying biochemical data demonstrate that SpoIVFB positions Pro-σ[K] at the enzyme active site through a *β*-sheet augmentation mechanism, and reveal key interactions between Pro-σ[K] and the interdomain linker connecting SpoIVFB transmembrane and CBS domains. The cryo-EM structure and molecular dynamics simulation reveal a plausible path for water to access the membrane-buried active site of SpoIVFB, and suggest a possible role of membrane lipids in facilitating substrate capture. These results provide key insight into how S2P intramembrane metalloproteases capture and position substrates for hydrolytic proteolysis within the hydrophobic interior of a lipid membrane.

Intramembrane proteolysis is a fundamental biochemical process that operates in organisms including Bacteria, Archaea, and Eukarya, and involves the hydrolytic cleavage of a peptide bond within the hydrophobic interior of a lipid membrane[1]. Enzymes that mediate intramembrane proteolysis generally fall into four categories: zinc metalloproteases exemplified by site-2 protease (S2P), the aspartic proteases exemplified by presenilin/signal peptide peptidase (SPP), the serine (Rhomboid) proteases, and the glutamic proteases exemplified by Ras-converting enzyme 1 (Rce1)[2]. Intramembrane proteolysis mediated by these enzymes plays an important role in biological processes ranging from environmental stress responses and sporulation in bacteria[3], to regulation of sterol homeostasis and progression of Alzheimer's disease in humans[1,4]. Among the four classes of intramembrane proteases, the S2P metalloproteases play important roles in activating transcription factors in all three domains of life[5].

The model organism *Bacillus subtilis* has served as one focal point of S2P intramembrane proteolysis research over the past two decades[6,7]. An intricate system of cellular checkpoints occurs

throughout *B. subtilis* sporulation, all of which are coordinated in a sophisticated and tightly regulated fashion[8]. One such checkpoint occurring late in *B. subtilis* endospore formation involves intramembrane proteolytic cleavage of the membrane-localized protein Pro-σ[K] into the mature transcription factor σ[K] (Fig. 1)[9]. Cleavage of Pro-σ[K] is mediated by the enzyme SpoIVFB, which is an S2P family intramembrane metalloprotease consisting of a six transmembrane (TM) helix catalytic domain, a 25-residue interdomain linker, and a C-terminal cystathionine-*β*-synthase (CBS) domain[10–12]. In the late stages of sporulation when SpoIVFB is expressed, the enzyme is held in an inactive state at the outer forespore membrane in a ternary complex with the inhibitory proteins SpoIVFA and BofA (Fig. 1B; **step 1**)[13,14]. Activation of SpoIVFB and subsequent Pro-σ[K] processing involves a series of upstream proteolytic events. These upstream events are initiated when the proteases SpoIVB and CtpB are translated in the interior of the forespore and secreted into the intermembrane space between the forespore and the mother cell cytoplasm, where they work in tandem to proteolytically cleave SpoIVFA in discrete steps[15–18]. Cleavage of

[1]Dept. of Biochemistry and Molecular Biology, Michigan State University, East Lansing, MI 48824, USA. [2]These authors contributed equally: Melanie A. Orlando, Hunter J. T. Pouillon, Saikat Mandal. ✉e-mail: orlandob@msu.edu

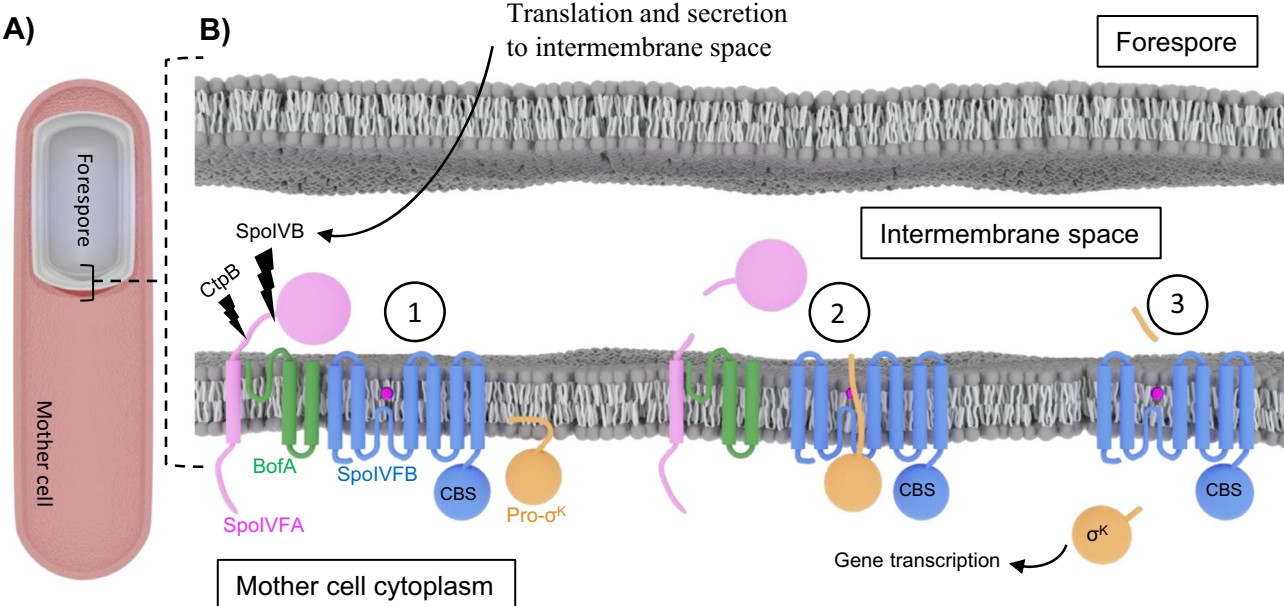

**Fig. 1 | Steps Involved in Pro-σ^K Cleavage by SpoIVFB. A** Sliced-view diagram of a sporulating *B. subtilis* cell. The mother cell is shown in red and the forespore is shown in gray. **B** Cartoon representation of the steps involved in relief of SpoIVFB inhibition and initiation of Pro-σ^K processing during *B. subtilis* sporulation. In the resting state (state 1) SpoIVFB is held inactive in a complex with BofA and SpoIVFA. The proteases SpoIVB and CtpB are produced in the forespore and secreted into the intermembrane space, where they proteolytically cleave SpoIVFA. Cleavage of SpoIVFA leads to relief of SpoIVFB inhibition, allowing the pro-sequence of Pro-σ^K to engage the active site of SpoIVFB (state 2). The proteolytic activity of SpoIVFB cleaves the pro-sequence of Pro-σ^K (state 3), allowing mature σ^K to diffuse from the vicinity of the outer forespore membrane and initiate further downstream gene transcription in the mother cell cytoplasm and progression of the sporulation process.

SpoIVFA by SpoIVB and CtpB serves to release the inhibition imposed upon SpoIVFB, and allows the substrate Pro-σ^K to engage the SpoIVFB active site (Fig. 1B; **step 2**)[19]. Finally, the zinc-metalloprotease activity of SpoIVFB results in cleavage of the N-terminal pro-sequence of Pro-σ^K within the interior of the lipid membrane, allowing the mature σ^K to diffuse from the outer forespore membrane into the mother cell cytoplasm, bind to RNA polymerase core subunits, and initiate transcription of genes involved in the final stages of *B. subtilis* spore maturation (Fig. 1B; **step 3**)[20,21]. This stepwise proteolytic mechanism is an elegant developmental checkpoint evolved by *B. subtilis* to ensure that transcription mediated by σ^K in the mother cell only occurs at the proper time during forespore development.

Despite extensive biochemical experiments that have illuminated the steps leading to Pro-σ^K cleavage outlined above, precise molecular insights into how SpoIVFB engages Pro-σ^K and catalyzes metal-mediated hydrolytic proteolysis within the hydrophobic interior of a lipid bilayer have remained elusive. Much of this uncertainty stems from the fact that a high-resolution structure of SpoIVFB has not been elucidated. SpoIVFB belongs to a large group of S2P intramembrane metalloproteases that is defined by the presence of a C-terminal CBS domain(s)[22]. Included in this group is the *Methanocaldococcus janaschii* site-2-protease (mjS2P), whose crystal structure provided a groundbreaking glimpse into the active site architecture of an S2P family intramembrane metalloprotease[2,23]. However, to crystallize mjS2P the C-terminal CBS domains were removed, and no substrate-engaged structure of the enzyme has since been determined. In fact, the natural endogenous substrate of mjS2P has yet to be identified. For these reasons a comprehensive understanding of the mechanism S2P intramembrane metalloproteases utilize for substrate capture and hydrolytic cleavage at a membrane-embedded active site, and the role of C-terminal CBS domain(s) in substrate binding/cleavage have remained enigmatic.

To unravel the molecular mechanism of Pro-σ^K cleavage mediated by SpoIVFB, we determine the cryo-electron microscopy (cryo-EM) structure of SpoIVFB bound to the substrate Pro-σ^K. The cryo-EM structures presented herein provide high-resolution molecular insights into substrate capture and positioning by a S2P intramembrane metalloprotease. The cryo-EM structures combined with extensive mutational analysis and protein crosslinking presented herein demonstrate the key protein-protein interactions that position Pro-σ^K for intramembrane proteolysis by SpoIVFB. Moreover, molecular dynamics simulations expose a plausible pathway for water to access the SpoIVFB active site that is positioned within the hydrophobic interior of the lipid bilayer. The results presented below lay the foundation to understand how diverse intramembrane metalloproteases engage their substrates to mediate hydrolysis within the lipid bilayer, and provide a template to begin further structural analysis of the critical checkpoints that regulate spore formation in *B. subtilis*.

## Results
### Purification and cryo-EM analysis of SpoIVFB:Pro-σ^K complexes
Previous studies have demonstrated that ~80–90% of Pro-σ^K can be processed into mature σ^K when the pro-protein is heterologously co-expressed with SpoIVFB in *Escherichia coli*[11,13]. However, some unprocessed Pro-σ^K remains detectable, and supplementation with zinc at the time of protein induction has no effect on the proportion of Pro-σ^K that is processed (Suppl. Fig. S1A). With this in mind, we initially began structural studies by co-expressing wild-type (WT) SpoIVFB and Pro-σ^K (both His-tagged) from the same plasmid but under the control of individual T7 promoters in *E. coli*. As expected, significant Pro-σ^K cleavage occurred upon co-expression of the two proteins, but after solubilizing membranes with detergent and purifying the His-tagged proteins through Co^{3+}-TALON affinity and size-exclusion chromatography, we consistently observed that SpoIVFB and full-length unprocessed Pro-σ^K co-purified as large protein complexes (Suppl. Fig. S1B, C). Despite extensive efforts, we were unable to reconstitute SpoIVFB enzymatic activity and Pro-σ^K cleavage in vitro using these detergent-solubilized preparations of WT SpoIVFB and Pro-σ^K (Suppl. Fig. S1F). An E44Q

mutation in the HEXXH motif of SpoIVFB completely abolishes the proteolytic activity of the enzyme[6,7]. When E44Q SpoIVFB was expressed and purified with Pro-σ$^K$ in *E. coli*, we obtained detergent-solubilized protein preparations that were similar on SDS-PAGE and size-exclusion chromatography (Suppl. Fig. S1D, E) to the preparations of WT SpoIVFB and Pro-σ$^K$. Thus, although WT SpoIVFB appears to display robust enzymatic activity in *E. coli*, this activity is lost after detergent solubilization and purification of WT SpoIVFB and Pro-σ$^K$.

To further investigate the overall architecture of detergent-solubilized SpoIVFB and Pro-σ$^K$ preparations, we pursued cryo-EM reconstructions of both WT and E44Q SpoIVFB co-purified with Pro-σ$^K$. Initial 2D classification of particles in preparations of WT SpoIVFB and Pro-σ$^K$ revealed a variety of oligomeric states, including dimers, trimers, and tetramers of SpoIVFB:Pro-σ$^K$ (Suppl. Fig. S2A, B). During 3D classification the dimeric and tetrameric species were most readily classified (Suppl. Fig. S2C) and attempts to reconstruct the SpoIVFB:Pro-σ$^K$ trimers were unsuccessful likely due to severe orientation bias of this particular species. We ultimately pursued a high-resolution reconstruction of the tetrameric species which resulted in a final cryo-EM map at 3.5 Å overall resolution (Fig. 2A **and** Suppl. Figs. S2c and S3). In this reconstruction the TM helices of SpoIVFB and the pro-sequence of Pro-σ$^K$ are particularly well resolved (Suppl. Fig. S3D, F), whereas the resolution is slightly lower in the cytosolic region of Pro-σ$^K$ and the CBS domain of SpoIVFB (Suppl. Fig. S3D, G).

In the WT SpoIVFB:Pro-σ$^K$ tetramers there are four copies of SpoIVFB:Pro-σ$^K$, but the complex only displays overall 2-fold symmetry (Fig. 2A **and** Suppl. Fig. S4A). Within the tetramer there are two interfaces between individual SpoIVFB:Pro-σ$^K$ monomers (Suppl. Fig S4A). Across interface 1 the SpoIVFB:Pro-σ$^K$ monomers are related by ~120° rotation, and across interface 2 the SpoIVFB:Pro-σ$^K$ monomers are related by ~60° rotation (Suppl. Fig S4A). Interestingly, an LMNG detergent molecule and a phospholipid-like density were clearly resolved in the cryo-EM reconstruction, and together the detergent and lipid appear to mediate much of the interaction between SpoIVFB

from individual SpoIVFB:Pro-σ$^K$ monomers across interface 1 (Suppl. Fig S4A, B). In contrast, most of the interaction across interface 2 is mediated by interaction between Pro-σ$^K$ from individual SpoIVFB:Pro-σ$^K$ monomers (Suppl. Fig S4C). In total, the physical interaction between monomers in the WT SpoIVFB:Pro-σ$^K$ tetramer is quite limited (~488Å$^2$ for interface 1, ~510Å$^2$ for interface 2), which likely makes for fragile interactions between SpoIVFB:Pro-σ$^K$ monomers that are largely mediated by lipid/detergent. As elaborated on below and in the discussion, we believe that the oligomeric arrangement of SpoIVFB:Pro-σ$^K$ monomers in the tetramer is likely an artifact of detergent solubilization, rather than a representation of an oligomeric state that is sampled in vivo.

In contrast to what was observed with WT SpoIVFB:Pro-σ$^K$ complexes, initial 2D and 3D classification of E44Q SpoIVFB:Pro-σ$^K$ particles showed a clear distribution of dimers and a lack of higher oligomeric species (Suppl. Fig. S5A, C). The final asymmetric cryo-EM reconstruction of dimeric E44Q SpoIVFB:Pro-σ$^K$ was obtained at an overall resolution of 3.9 Å, with the TM helices of SpoIVFB and the pro-sequence of Pro-σ$^K$ particularly well resolved (Suppl. Fig. S6A, G). By comparing the final cryo-EM reconstructions of WT and E44Q SpoIVFB:Pro-σ$^K$ complexes it becomes readily apparent that E44Q SpoIVFB:Pro-σ$^K$ is approximately one half the size of the tetrameric WT complex (Fig. 2B), and contains two monomers that are related to one another by ~120° rotation similar to that observed across interface 1 in the WT complexes (Suppl. Fig. S4A). Aside from the difference in oligomeric state, which we believe arises simply from variations between protein preparations and fragile monomer interfaces, the final structures of WT and E44Q SpoIVFB:Pro-σ$^K$ monomers are virtually superimposable (RMSD ~ 0.5 Å). For this reason, we refer to WT SpoIVFB:Pro-σ$^K$ in the following presentation of results due to the slightly higher overall resolution obtained for this complex.

The structure of SpoIVFB provides insight into the configuration of the C-terminal CBS domain that was truncated to obtain a previous structure of mjS2P[23], as well as the interdomain linker that connects SpoIVFB TM and CBS domains (Fig. 3A). A sharp kink is observed at the

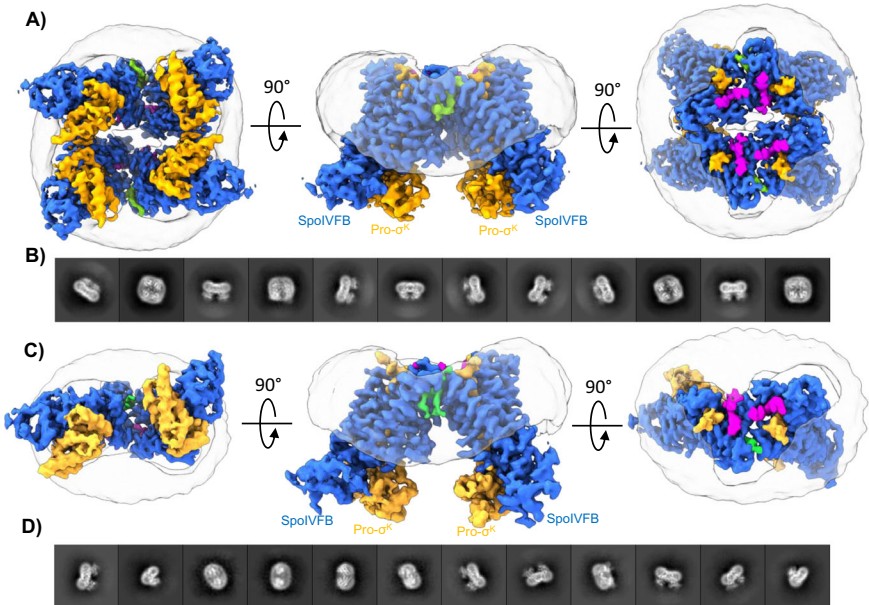

**Fig. 2 | Cryo-EM Structure of SpoIVFB:Pro-σ$^K$ Complexes. A** Three rotated views of the tetrameric cryo-EM structure of WT SpoIVFB:Pro-σ$^K$. SpoIVFB is colored blue, Pro-σ$^K$ is colored orange, LMNG detergent is colored magenta, a lipid-like density at the interface between monomers is colored green, and the detergent micelle is shown in light transparent gray. **B** Selected 2D averages of the tetrameric WT SpoIVFB:Pro-σ$^K$ complex showing four copies of WT SpoIVFB:Pro-σ$^K$ arranged with overall 2-fold symmetry. **C** Three rotated views of the dimeric cryo-EM structure of E44Q SpoIVFB:Pro-σ$^K$. Coloring is the same as in (**A**). **D** Selected 2D averages of the dimeric E44Q SpoIVFB:Pro-σ$^K$ complex showing two copies of E44Q SpoIVFB:Pro-σ$^K$ arranged with no symmetry.

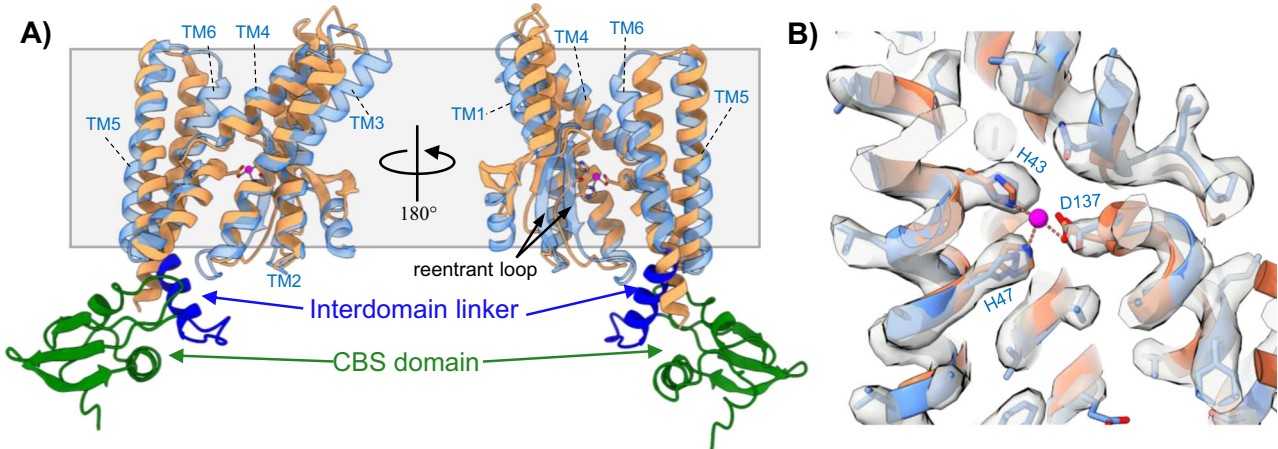

**Fig. 3 | Structure of the SpoIVFB Intramembrane Protease. A** Comparison of the SpoIVFB structure with the previous crystal structure of mjS2P in an open conformation (PDB: 3B4R). SpoIVFB is colored with the TM helices in transparent light blue, the interdomain linker in dark blue, and the CBS domain in green. mjS2P is colored in orange. The gray box delineates the approximate boundary of the lipid

membrane. **B** Close-up view of the SpoIVFB and mjS2P zinc-binding site. SpoIVFB is colored blue, mjS2P is colored orange, and the zinc identified in the mjS2P structure is colored magenta. The cryo-EM map of WT SpoIVFB:Pro-σ$^K$ is shown as a transparent gray surface. The cryo-EM map demonstrates that no zinc ion was co-purified with WT SpoIVFB:Pro-σ$^K$. Potential zinc-coordinating residue numbers are from SpoIVFB.

end of SpoIVFB TM6 which causes the interdomain linker to orient at an angle relative to the lipid membrane. Directly after the interdomain linker is the C-terminal CBS domain, which is positioned underneath the cytosolic regions of SpoIVFB TM4-6 (Fig. 3A). The topology of TM helices in SpoIVFB and mjS2P is the same, but the pitch of some TM helices is different (RMSD ~ 3.5 Å overall). Both proteins contain a membrane reentrant loop located between TM2 and TM3 composed of antiparallel β-strands (Fig. 3A). The reentrant loop spans roughly halfway across the lipid membrane and is positioned directly in front of the HEXXH zinc-binding motif previously identified in mjS2P[23]. While the orientation of zinc-coordinating residues H43, H47, and D137 of SpoIVFB are conserved with those of mjS2P, the cryo-EM map of WT SpoIVFB:Pro-σ$^K$ does not show any appreciable density for an ion being coordinated by these residues (Fig. 3B). This lack of density was consistent in the cryo-EM maps of both WT and E44Q SpoIVFB:Pro-σ$^K$, demonstrating that SpoIVFB:Pro-σ$^K$ complexes fail to co-purify with the Zn$^{2+}$ ion that is essential for supporting Pro-σ$^K$ cleavage. Thus, the cryo-EM maps provide an explanation for why WT SpoIVFB consistently co-purifies with full-length unprocessed Pro-σ$^K$ despite displaying robust proteolytic activity in *E. coli*. Attempts to reconstitute in vitro proteolytic cleavage of Pro-σ$^K$ by supplementing purified WT SpoIVFB:Pro-σ$^K$ complexes with Zn$^{2+}$ (by adding Zn-acetate or Zn-chloride) have thus far been unsuccessful. Although the purified WT SpoIVFB:Pro-σ$^K$ complexes are catalytically inert due to lack of a critical Zn$^{2+}$ ion, important mechanistic information about the binding of Pro-σ$^K$ to SpoIVFB can still be gleaned from the cryo-EM structures.

### Engagement of the pro-sequence of Pro-σ$^K$ in the SpoIVFB active site

In the binding pose captured in our cryo-EM structures the pro-sequence of Pro-σ$^K$ extends across the entire lipid bilayer, with the N-terminus pointing into the intermembrane space and the C-terminal alpha-helical subdomain of Pro-σ$^K$ in the mother cell cytoplasm (Fig. 4A, B). The pro-sequence of Pro-σ$^K$ threads adjacent to the HEXXH zinc-binding motif of SpoIVFB and forms an additional β-strand with the SpoIVFB membrane reentrant loop (Fig. 4A). Our cryo-EM structures thus provide direct evidence of a β-sheet augmentation mechanism of substrate engagement which has previously been proposed for distantly related S2P intramembrane metalloproteases such as *E. coli* RseP[24]. Interestingly, an LMNG detergent molecule was observed directly over the SpoIVFB membrane reentrant loop and

packed against the N-terminal region of the pro-sequence of Pro-σ$^K$, possibly mimicking a membrane phospholipid (Fig. 4A, B).

Previous investigations have revealed that the pro-sequence of Pro-σ$^K$ is cleaved between residues S21 and Y22[25,26] (Fig. 4C), and in our cryo-EM structures these two residues are positioned adjacent to the SpoIVFB HEXXH zinc-binding motif in a position that appears primed for catalysis (Fig. 4B). In this position the carbonyl oxygen in the peptide backbone between S21 and Y22 of Pro-σ$^K$ points towards the HEXXH motif where zinc would be coordinated (Suppl. Fig S7A). The orientation of this backbone carbonyl is consistent with a proposed mechanism where the coordinated Zn$^{2+}$ ion would stabilize the oxyanion intermediate that develops on the backbone carbonyl during peptide hydrolysis[27] (Suppl. Fig S7B). By adding a β-strand to the SpoIVFB membrane reentrant loop β-sheet, the pro-sequence of Pro-σ$^K$ observed in our cryo-EM structures appears to be ideally positioned for intramembrane proteolysis if the enzyme preparations contained the catalytic Zn$^{2+}$ ion and water could reach the vicinity of the enzyme active site.

In order to verify that the position of the pro-sequence observed in the cryo-EM structure represents a state that precedes initiation of the metal-mediated hydrolysis reaction, we took advantage of the robust enzymatic activity of SpoIVFB that is observed in *E. coli*. The β-sheet structure of the pro-sequence of Pro-σ$^K$ causes successive residues in the sequence to alternate between pointing toward or away from the SpoIVFB active site (Fig. 4B). For example, V20, Y22, and K24 of Pro-σ$^K$ point away from the zinc-binding site, whereas L19, S21, and V23 point inward toward the zinc-binding site (Fig. 4B). We reasoned that replacing residues that face toward the zinc-binding site with a bulkier side chain would inhibit proper engagement of the pro-sequence with SpoIVFB due to steric interference, whereas similar bulky replacements in residues that orient away from the enzyme active site should have less effect on proteolysis. To investigate this hypothesis, we performed tryptophan scanning mutagenesis along the pro-sequence of Pro-σ$^K$ and probed the ability of these variants to be proteolytically processed in *E. coli* (Fig. 4D & E). Tryptophan variants of L16, V17, and F18 had little to no effect on Pro-σ$^K$ cleavage, likely because they are positioned slightly above the β-sheet structure of the pro-sequence and membrane reentrant loop where bulky residues are more easily tolerated. In contrast, tryptophan variants of S21 and V23 completely abolished Pro-σ$^K$ cleavage, likely because a tryptophan residue in these positions leads to steric clash with residues surrounding the SpoIVFB active site and mis-positioning of the pro-

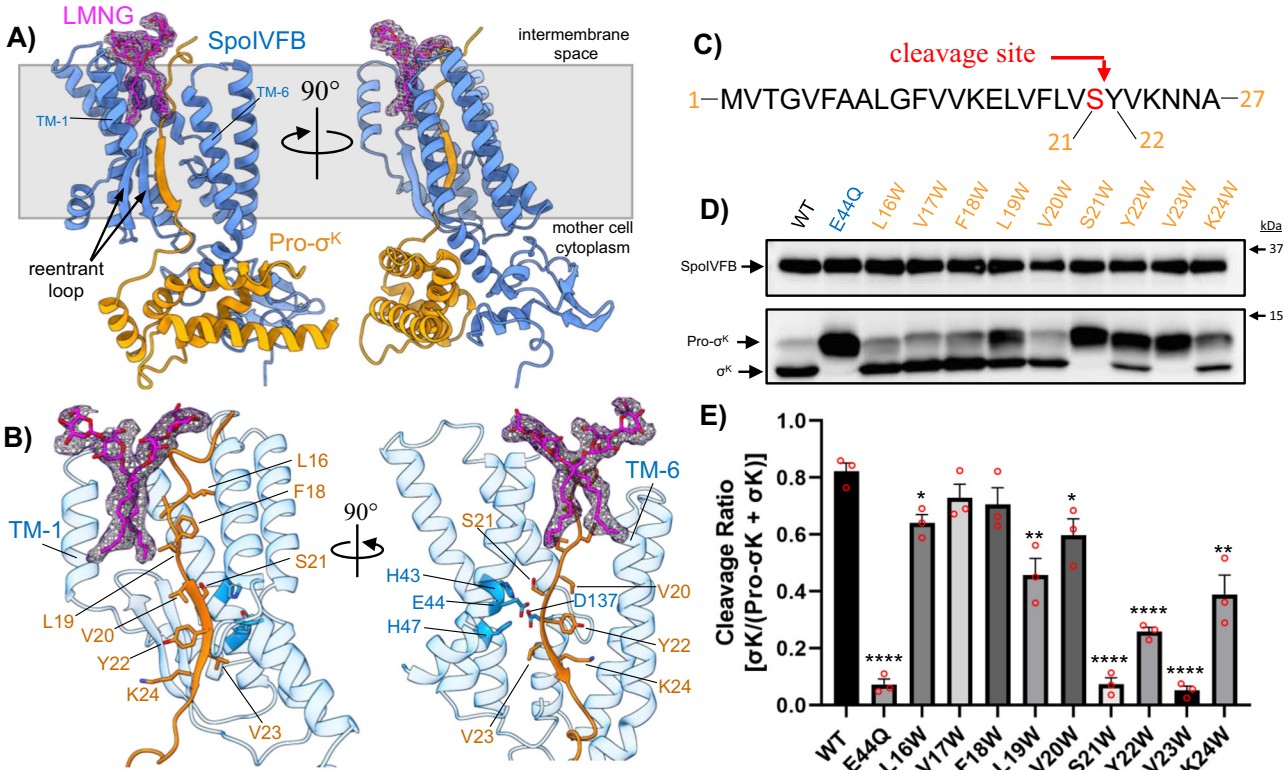

**Fig. 4 | Binding of the Pro-sequence of Pro-σ^K to SpoIVFB. A** Two views of a single monomer of the WT SpoIVFB:Pro-σ^K complex. SpoIVFB is colored blue, Pro-σ^K is colored orange, LMNG detergent is colored magenta with the corresponding cryo-EM map shown in magenta mesh. The gray box approximates the lipid membrane. The pro-sequence of Pro-σ^K engages the reentrant loop of SpoIVFB through b-sheet augmentation. The LMNG detergent molecule is positioned between TM1 and TM6, directly over the reentrant loop. **B** Close-up view of the Pro-σ^K pro-sequence engaging SpoIVFB through b-sheet augmentation. Individual residues of the pro-sequence are indicated. The three zinc-coordinating residues and catalytic E44 of SpoIVFB are highlighted in darker blue. The reentrant loop of SpoIVFB and TM1 are deleted from the image on the right to reveal the enzyme active site. **C** Schematic showing the amino acid sequence at the N-terminus of Pro-σ^K. The site of SpoIVFB-mediated cleavage is highlighted in red. **D** Anti-pentaHis western blot showing Pro-

σ^K cleavage by SpoIVFB variants expressed in *E. coli*. The bands corresponding to SpoIVFB, Pro-σ^K, and processed σ^K are indicated to the left. Approximate locations of molecular weight markers are shown to the right. **E** Densitometry-based quantification of Pro-σ^K cleavage for the variants shown in (**D**). Bars represent the average cleavage ratio ([σ^K]/[Pro-σ^K + σ^K]) calculated from three (n = 3) independent experiments, with error bars representing standard error of the mean (SEM). Red circles indicate individual data points from the triplicate measurements. Significance from an unpaired two-sided *t*-test between each variant and WT is indicated with asterisks (* $P \leq 0.05$, ** $P \leq 0.01$, *** $P \leq 0.001$, **** $P \leq 0.0001$). P values for each variant are E44Q = 0.001, L16W = 0.0119, V17W = 0.1737, F18W = 0.1565, L19W = 0.0051, V20W = 0.0254, S21W = 0.0001, Y22W = 0.0001, V23W = 0.0001, K24W = 0.0041.

sequence for enzymatic cleavage. Variants of residues on either side of S21 or V23 (V20W, Y22W, and K24W) still supported cleavage of Pro-σ^K, albeit at slightly lower levels than observed with the WT sequence (Fig. 4D, E). In total, the results of the tryptophan scanning mutagenesis agree with our predictions based on the β-sheet architecture of the pro-sequence observed in the cryo-EM structure, lending support to the notion that the cryo-EM structure represents a state encountered just before initiation of intramembrane proteolysis.

**Interaction between Pro-σ^K and SpoIVFB in the mother cell cytoplasm**

Pro-σ^K contains an alpha-helical subdomain that is typical of other sigma transcription factors[28]. In our cryo-EM reconstructions this alpha-helical subdomain extends beneath the plane of the lipid membrane and into the mother cell cytoplasm (Fig. 4A). Just beneath the membrane plane the alpha-helical subdomain of Pro-σ^K interacts with the interdomain linker that connects the TM and CBS domains of SpoIVFB (Figs. 3A and 5A). Pro-σ^K packs against the interdomain linker through helix-helix interactions that are largely hydrophobic in nature (Fig. 5A). The CBS domain of SpoIVFB is oriented to the opposite side of the interdomain linker relative to Pro-σ^K, and only limited interaction between the CBS domain and Pro-σ^K is observed. Interestingly, the CBS domain seems to be held in place by an electrostatic interaction

between R244 in the CBS domain and E83 in the TM domain of SpoIVFB (Fig. 5A). Although both R244 and E83 are largely conserved across orthologs of SpoVFB, we found that mutating either of these residues to alanine had no effect on the ability of SpoIVFB to process Pro-σ^K when co-expressed in *E. coli* (Suppl. Fig. S7C, D).

To investigate the importance of residues along the SpoIVFB interdomain linker in mediating interaction with Pro-σ^K we performed alanine scanning mutagenesis along the linker and probed the ability of these variants to process Pro-σ^K in an *E. coli*-based cleavage assay (Fig. 5B, C). It should be noted that a similar approach was attempted previously, and residues H206, F209, and R213 of SpoIVFB were found to be essential for Pro-σ^K processing[11]. These three residues in the SpoIVFB interdomain linker point away from Pro-σ^K and towards the CBS domain in our cryo-EM structure (Fig. 5A). In the previous experiments SpoIVFB and Pro-σ^K were expressed from separate plasmids simultaneously in *E. coli*, and our current strategy of co-expressing SpoIVFB and Pro-σ^K from the same plasmid with individual T7 promoters consistently results in much more robust cleavage of Pro-σ^K. Using the single plasmid system we found that the R213A variant had slightly impaired ability to process Pro-σ^K, and H206A and F209A variants processed Pro-σ^K comparable to WT SpoIVFB (Fig. 5B, C). We scanned several other alanine variants along the length of the interdomain linker and failed to identify any single point

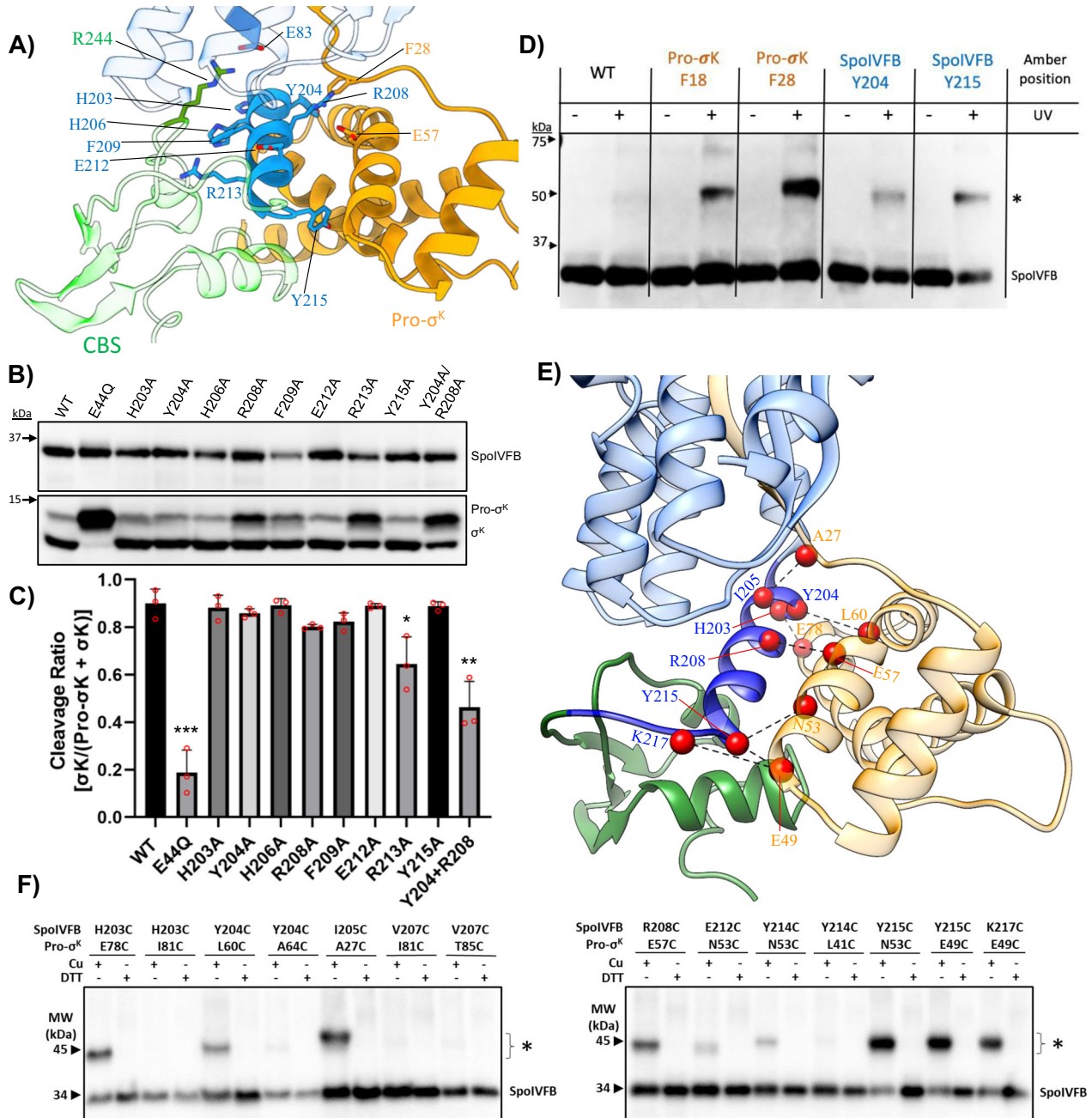

**Fig. 5 | Interaction Between Pro-σ^K and the SpoIVFB Interdomain Linker. A** View of the interaction between the SpoIVFB interdomain linker (dark blue) and the alpha-helical cytosolic domain of Pro-σ^K (orange). The CBS domain of SpoIVFB is colored transparent green. **B** Anti-pentaHis western blot showing Pro-σ^K cleavage by SpoIVFB interdomain linker variants expressed in *E. coli*. Bands corresponding to SpoIVFB, Pro-σ^K, and processed σ^K are indicated to the right. Approximate locations of molecular weight markers are shown to the left. **C** Densitometry-based quantification of Pro-σ^K cleavage by SpoIVFB variants in the interdomain linker from (**B**). Bars represent the average cleavage ratio ([σ^K]/[Pro-σ^K + σ^K]) calculated from three (*n* = 3) independent experiments, with error bars representing standard error of the mean (SEM). Red circles indicate individual data points from the triplicate measurements. Significance from an unpaired two-sided *t*-test between each variant and WT is indicated with asterisks (* $P \leq 0.05$, ** $P \leq 0.01$, *** $P \leq 0.001$, **** $P \leq 0.0001$). P values for each variant are E44Q = 0.004, H203A = 0.7336, Y204A = 0.3242, H206A = 0.8544, R208A = 0.0520, F209A = 0.1382, E212A = 0.7979,

R213A = 0.270, Y215A = 0.7856, Y204A/R208A = 0.0037. **D** Anti-pentaHis western blot showing in vivo photocrosslinking between SpoIVFB and Pro-σ^K. The position of amber stop codons in SpoIVFB or Pro-σ^K is indicated at the top. Crosslinked SpoIVFB species are indicated with an asterisk. Approximate molecular weight markers are shown to the left. The experiment was repeated once. **E** Close-up view of the interface between SpoIVFB and Pro-s^K. Pairs of residues that showed strong disulfide crosslinking (**F**) are shown with red spheres and connecting dotted black lines. **F** Anti-FLAG western blots showing in vivo disulfide crosslinking between single-cysteine variants of SpoIVFB and Pro-s^K. The position of cysteine residues in SpoIVFB and Pro-s^K is indicated at the top, as is treatment with oxidant Cu^{2+}(phenanthroline)$_3$ (Cu) (1 mM) or reductant dithiothreitol (DTT) (100 mM) to promote or inhibit disulfide crosslinking, respectively. Crosslinked SpoIVFB species are indicated with an asterisk. The position of migration of protein molecular weight (MW) markers is shown. The experiment was repeated once.

mutation that led to statistically significant alterations in Pro-σ$^K$ processing. However, a double alanine variant Y204A/R208A led to a significant reduction in Pro-σ$^K$ processing (Fig. 5B, C). Both Y204 and R208 point towards the alpha-helical subdomain of Pro-σ$^K$, with R208 forming an electrostatic interaction with E57 of Pro-σ$^K$ (Fig. 5A). The collective results of alanine scanning along the SpoIVFB interdomain linker suggest that no single residue is critical for interaction with Pro-σ$^K$, but rather that hydrophobic and electrostatic interactions along the helix-helix packing interface collectively support interaction with Pro-σ$^K$.

To further validate the interaction interfaces between Pro-σ$^K$ and SpoIVFB that we observe in the cryo-EM structure, we turned to two different methods of in vivo crosslinking. In the first approach we incorporated the UV reactive unnatural amino acid p-azidophenylalanine (pAzF) at different positions in SpoIVFB or Pro-σ$^K$. We chose to incorporate pAzF at two positions in Pro-σ$^K$ (F18 or F28) predicted to be proximal to SpoIVFB (Figs. 4B & 5A), as well as two positions in the SpoIVFB interdomain linker (Y204 or Y215) predicted to interact with the alpha-helical subdomain of Pro-σ$^K$ (Fig. 5A). Following UV exposure of *E. coli* cells expressing these variants we observed a higher molecular weight species indicative of covalent crosslinking between SpoIVFB and Pro-σ$^K$ (Fig. 5D). Incorporation of pAzF into regions of the Pro-σ$^K$ pro-sequence led to significantly stronger crosslinking than when pAzF was incorporated into the SpoIVFB interdomain linker. The slightly bulkier side-chain of pAzF is likely to be more easily tolerated in the pro-sequence of Pro-σ$^K$ (similar to bulky residue additions in the tryptophan scanning mutagenesis above), whereas introducing a bulkier side-chain at the interface of the SpoIVFB interdomain linker and alpha-helical subdomain of Pro-σ$^K$ may lead to weaker interaction in this region.

In addition to UV photocrosslinking we also performed site-directed disulfide crosslinking to validate the interface between SpoIVFB and Pro-σ$^K$ observed in the cryo-EM structure. Fourteen different pairs of cysteine residues were incorporated along the SpoIVFB interdomain linker and the alpha-helical subdomain of Pro-σ$^K$ (Suppl. Fig. S8A), and Cu$^{2+}$(phenanthroline)$_3$ was used to induce oxidative disulfide formation in intact *E. coli* expressing these variants. Cysteine pairs that formed disulfide crosslinks were identified along the entirety of the SpoIVFB interdomain linker (Fig. 5E, F and Suppl. Fig. S8). The strongest among these crosslinks appeared at the C-terminal end of the interdomain linker just before the CBS domain, including crosslinks between Y215 of SpoIVFB and N53 or E49 of Pro-σ$^K$ (Fig. 5E, F and Suppl. Fig. S8B, E). A strong crosslink was also observed between I205 at the N-terminal region of the SpoIVFB interdomain linker and A27 which lies just beyond the region of Pro-σ$^K$ that forms a β-sheet with the membrane reentrant loop of SpoIVFB (Fig. 5E, F). Collectively, the results of disulfide crosslinking and UV photocrosslinking confirm that the interaction observed between SpoIVFB and Pro-σ$^K$ in the cryo-EM structure are representative of the protein-protein interactions that occur when the complex resides within a lipid bilayer in intact *E. coli*.

### Access of water and lipids near the SpoIVFB active site

According to the proposed mechanism for SpoIVFB catalyzed intramembrane proteolysis (Suppl. Fig. S7B), a water molecule needs to penetrate approximately halfway across the hydrophobic interior of the lipid membrane and into the vicinity of the zinc-binding site (to be activated by catalytic SpoIVFB residue E44). To investigate how water may access the SpoIVFB active site we performed all-atom molecular dynamics (MD) simulations on the cryo-EM structure of the WT SpoIVFB:Pro-σ$^K$ complex that was embedded in a lipid bilayer designed to mimic the lipid composition previously found in *B. subtilis* membranes (Suppl. Fig. S9A, B)[29]. Four replicate simulations were performed on a monomer of the SpoIVFB:Pro-σ$^K$ complex which lacked a zinc ion, similar to what was observed in the cryo-EM structure. Throughout 250 ns of unrestrained simulation for each replicate, the

SpoIVFB:Pro-σ$^K$ complex remained remarkably stable (Suppl. Fig. S9C–F), suggesting that the conformation of SpoIVFB:Pro-σ$^K$ observed in the cryo-EM structure is consistent with a conformation that would be adopted in a lipid membrane.

During the simulations water penetrated from the environment of the mother cell cytoplasm up through the base of the SpoIVFB TM helices in the region where E83 interacts with R244 in the CBS domain (Figs. 5A and 6A, B). Water entering through this cytoplasmic opening in the TM helices traversed nearly halfway across the lipid bilayer and filled the region that would be occupied by the catalytic zinc ion and SpoIVFB E44 side chain (Fig. 6A–D) just behind the pro-sequence of Pro-σ$^K$. Calculating the number of water molecules within 7 Å distance of the catalytic E44 residue over time reveals a similar pattern of rapid water influx from the mother cell cytoplasm to the general SpoIVFB active site region (Suppl. Fig. S9G–L and Suppl. Movie 1). Our MD simulations present a plausible scenario whereby water from the mother cell cytoplasm can penetrate between the TM helices of SpoIVFB to reach the level of the zinc-binding site where it is required to facilitate hydrolytic scission of the Pro-σ$^K$ peptide backbone. Additionally, water from the mother cell cytoplasm filled the space directly in front of the membrane reentrant loop and the pro-sequence of Pro-σ$^K$ (Fig. 6A, C). Water partitioning near the front of the membrane reentrant loop effectively thins the lipid membrane in the area of the SpoIVFB active site entrance between TM1 and TM6, which may facilitate the entry of Pro-σ$^K$ into the active site binding cleft.

In addition to the penetration of water into the SpoIVFB active site, our MD simulations revealed another interesting insight into the role of the lipid bilayer in possibly facilitating Pro-σ$^K$ processing. In each of the replicate simulations a lipid molecule(s) moved near or directly over the SpoIVFB membrane reentrant loop and remained in this position through the entire 250 ns simulation (Fig. 6E). In three of the replicate simulations the lipid happened to be a DAG, which lacks a phospholipid headgroup. In the remaining replicate simulation the lipid was a POPG. By interacting with SpoIVFB directly over the membrane reentrant loop, the lipid closely mimics the binding pose for an LMNG detergent molecule that was observed in the cryo-EM maps (Fig. 6E). Both the lipid and LMNG are positioned over the membrane reentrant loop with a single acyl-chain extending down over either side of the loop, and the N-terminal region of the Pro-σ$^K$ pro-sequence packed closely against the bound detergent/lipid (Fig. 6E). At the apex of the membrane reentrant loop and just beneath the lipid/detergent is SpoIVFB residue F66 (Fig. 6E), which has previously been implicated in regulating a closed-open transition of SpoIVFB to allow for substrate access into the active site cleft[19]. In this position above the membrane reentrant loop and above residue F66, the lipid/LMNG molecule effectively seals off the cleft between SpoIVFB TM1 and TM6, and forces the N-terminal region of Pro-σ$^K$ to point vertically into the intermembrane space (Fig. 6E). Analyzing the surface of the cryo-EM structure reveals a hydrophobic cleft behind the membrane reentrant loop and near TM1 and TM2 (Fig. 6F). The acyl chains of the LMNG molecule in the cryo-EM structure and the lipid molecules in simulations both occupy this hydrophobic pocket. The fact that DAG and POPG both traversed within this region in replicate simulations, and that LMNG is bound stably enough in this region to be observed by cryo-EM, both suggest that hydrophobic interactions between the acyl chains of these molecules and the hydrophobic pocket of SpoIVFB are major drivers of interaction rather than specific lipid headgroup interaction. Both the cryo-EM structure and MD simulations seem to support a potential role of membrane lipids in helping to stabilize the interaction and conformation of Pro-σ$^K$ bound to SpoIVFB, possibly facilitating the overall proteolytic cleavage mechanism.

## Discussion

The cryo-EM structure of SpoIVFB bound to the proteolytic substrate Pro-σ$^K$ provides long-sought structural insight into the substrate

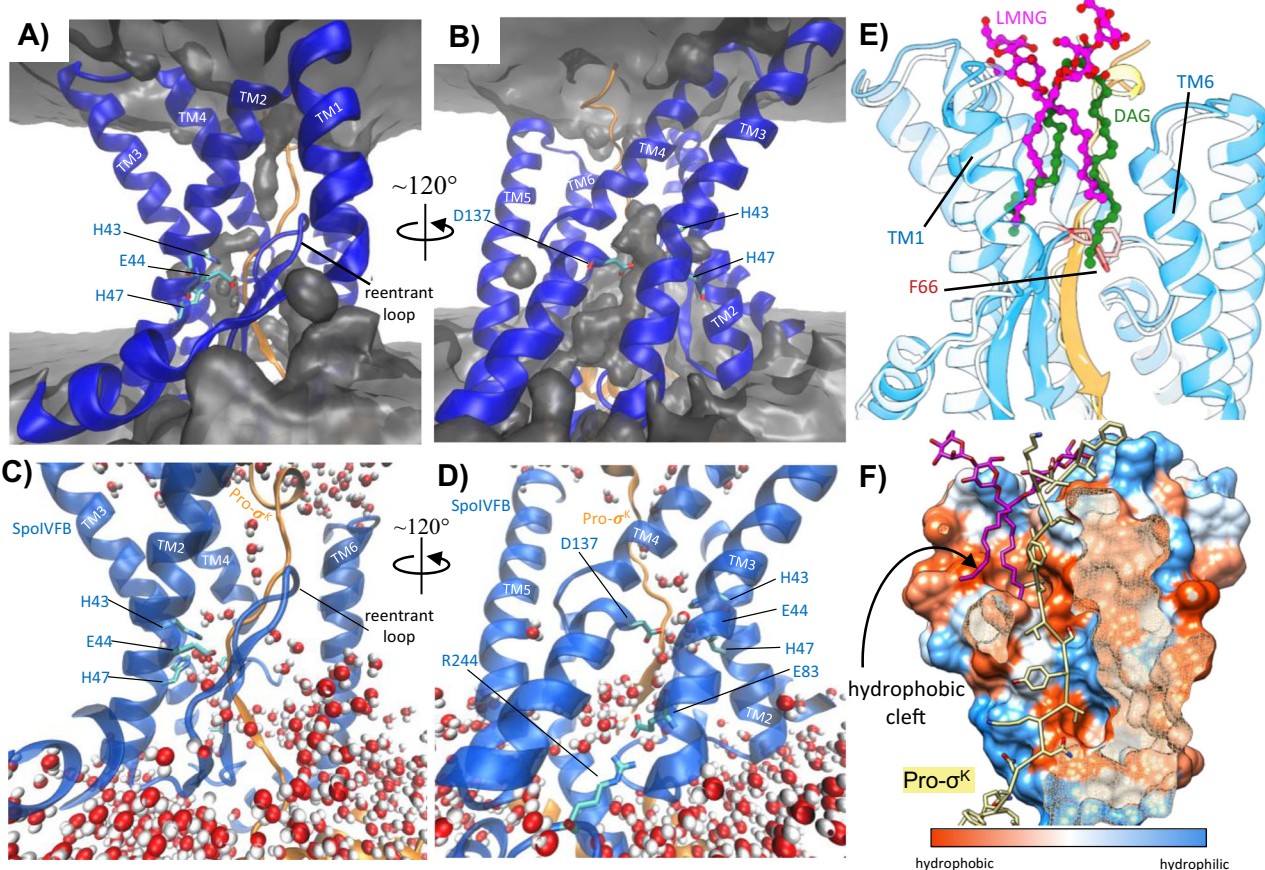

**Fig. 6 | Molecular Dynamics Analysis of the SpoIVFB:Pro-σ^K Complex.**
**A**, **B** Rotated views of the SpoIVFB:Pro-σ^K complex (SpoIVFB is colored blue, Pro-σ^K is colored orange). A map showing the average occupancy of water calculated across one replicate 250 ns simulation is shown as a gray surface. Water fills the area in front of the membrane reentrant loop (**A**) and in the region of the zinc binding site (**B**). **C**, **D** Zoomed in views of the SpoIVFB active site from the final frame of one replicate MD simulation showing positions of waters at the end of the simulation. **E** Overlay of the SpoIVFB:Pro-σ^K cryo-EM structure (dark blue and orange) with the final frame from one replicate 250 ns MD simulation (light blue and yellow). The LMNG detergent molecule observed in the cryo-EM structure is shown in magenta ball and sticks, and a DAG lipid observed in one replicate MD simulation is shown in green ball and sticks. The acyl chains of LMNG mimic the acyl chains of the DAG lipid. F66 at the apex of the membrane reentrant loop is shown for clarity. **F** Surface representation of SpoIVFB with the surface colored according to hydrophobicity. The pro-sequence of Pro-σ^K is shown as yellow sticks, and the LMNG in the cryo-EM structure is shown in magenta sticks. The acyl chains of the detergent molecule stick into a hydrophobic pocket located behind the membrane reentrant loop.

engagement mechanism of S2P intramembrane metalloproteases in general, as well as a key checkpoint in the late stages of *B. subtilis* sporulation. Our cryo-EM structures demonstrate that capture of Pro-σ^K in the SpoIVFB active site occurs largely through β-sheet augmentation of the SpoIVFB membrane reentrant loop. Once Pro-σ^K engages with SpoIVFB in this region, the pro-sequence of the substrate is extended in a β-sheet conformation that is poised to facilitate hydrolytic cleavage at the backbone carbonyl that resides between Pro-σ^K residues S21 and Y22 (Suppl. Figs. S7A, B and S10A, E). A similar mechanism of β-strand addition was also suggested to mediate substrate engagement by the distantly related S2P protease RseP in *E. coli* (Suppl. Fig. S10B, F)[24,30]. A crystal structure of the Rhomboid protease GlpG bound to a mechanism-based inhibitor that mimics the natural substrate also showed the substrate mimic in an extended β-sheet conformation in the enzyme active site (Suppl. Fig. S10C, G)[31]. The cryo-EM structure of the human aspartyl intramembrane protease γ-secretase bound to the substrate Notch also revealed a mechanism of β-sheet augmentation in substrate capture (Suppl. Fig. S10D, H)[32,33]. Interestingly, even in intramembrane proteases that are processive and catalyze multiple successive cleavage reactions on a single substrate (i.e., gamma-secretase cleaving APP/Aβ), successive reengagement of the substrate through beta-sheet augmentation appears to position successive cleavage sites for proteolysis (Suppl. Fig. S10I, K).

Thus, the SpoIVFB:Pro-σ^K structures presented here lend support to the notion that capture of proteolytic substrates in an extended β-sheet conformation is a general mechanism operating broadly across several classes of intramembrane proteases.

The role of the CBS domain in SpoIVFB and related S2P proteases has remained enigmatic[11,19,34,35]. Whereas the tandem CBS domains of mjS2P were completely dispensable for in vitro cleavage of the surrogate substrate CED-9 from *Caenorhabditis elegans*[23], an investigation in sporulating *B. subtilis* demonstrated that the SpoIVFB CBS domain promotes localization of Pro-σ^K to the outer forespore membrane, but is not essential for Pro-σ^K processing and progression of sporulation[19]. However, when both the CBS domain and the interdomain linker between the CBS and TM domains of SpoIVFB were removed, this truncated construct was unable to support Pro-σ^K processing and sporulation. These and other results suggested that the interdomain linker is a major determinant for mediating interaction between Pro-σ^K and SpoIVFB, and the CBS domain likely aids in proper positioning of the interdomain linker to receive Pro-σ^K[11,19]. Such a scenario is borne out in our cryo-EM structure with Pro-σ^K interacting extensively with the interdomain linker, and much less so with the CBS domain (Fig. 5A). Sequence conservation among SpoIVFB orthologs indicates that the C-terminal region of the interdomain linker is quite well conserved (Suppl. Fig. S7C) and we observed strong disulfide crosslinks between

this region and Pro-σ$^K$ (Fig. 5E, F), suggesting that interaction between the C-terminal region of the interdomain linker and substrate is a conserved feature of substrate engagement for SpoIVFB and its orthologs. The location of the CBS domain beneath the TM region of SpoIVFB likely aids in positioning the interdomain linker to support binding and processing of Pro-σ$^K$. CBS domains commonly appear in tandem repeats and undergo dimerization in response to small molecules such as ATP or other cofactors[36]. From our cryo-EM structures it is difficult to postulate how dimerization of CBS domains from different SpoIVFB monomers would lead to conformational shifts across the enzyme structure. The contribution of the CBS domain to Pro-σ$^K$ processing in the context of the full SpoIVFA:BofA:SpoIVFB complex will require further investigation.

For S2P proteases to catalyze intramembrane proteolysis, water must be able to penetrate nearly halfway across the hydrophobic interior of the lipid bilayer and access the general vicinity of the enzyme active site and scissile peptide bond of the substrate. The previous crystal structure of mjS2P revealed a potential hydrophilic tunnel that spanned from the cytosol to the zinc active site[23]. However, this tunnel was observed in an alternate closed conformation of mjS2P that is not observed here for substrate engaged SpoIVFB. Moreover, engagement of Pro-σ$^K$ significantly limits potential routes for water to access the enzyme active site. From our MD analysis it appears that a pathway for water to reach the SpoIVFB active site is formed at the base of the SpoIVFB TM helices in the mother cell cytoplasm (Fig. 6A, B). By entering through this opening water was able to traverse behind the bound Pro-σ$^K$ substrate and reach the vicinity of the enzyme active site and residue E44Q which is proposed to activate a water for nucleophilic attack of a backbone carbonyl (Suppl. Fig. S7B). Thus, even after an S2P intramembrane protease has engaged a proteolytic substrate it appears as though water can still traverse through a hydrophilic tunnel to the enzyme active site.

While the data presented here provide key insights into the mechanisms that lead to intramembrane proteolysis of Pro-σ$^K$, they also raise several important points for consideration. In both our WT and E44Q SpoIVFB:Pro-σ$^K$ preparations the purified protein is devoid of the catalytic zinc ion. We have thus far been unable to recover enzymatic activity by supplementing zinc back to the purified WT complex. Why then does SpoIVFB seem to display robust enzymatic activity in intact E. coli? It is possible that the catalytic zinc ion is lost during protein purification, and once Pro-σ$^K$ binds to a zinc-depleted SpoIVFB the pro-sequence blocks the zinc-coordinating residues and prevents future complexation with exogenously supplied zinc. Although water appears to penetrate from the mother cell cytosol to the zinc active site even when Pro-σ$^K$ is bound to SpoIVFB (Fig. 6A, B), it is unclear if this pathway would also support diffusion of a zinc ion to the active site. Another possible explanation is that the majority of SpoIVFB that is expressed in E. coli indeed lacks zinc, and the proteolytic activity observed in these cells arises from a small fraction of SpoIVFB that contains the catalytically essential ion. While SpoIVFB cleavage of Pro-σ$^K$ has been reconstituted in vitro[34], further work is warranted to develop a more robust proteolysis assay to probe SpoIVFB activity with purified components. Complicating such an endeavor is the fact that SpoIVFB is refractory to mild detergent solubilization when expressed alone without Pro-σ$^{K37}$, which makes isolation of active SpoIVFB particularly challenging.

Another point for consideration is the oligomeric state of SpoIVFB. In this work we identified dimeric, trimeric, and tetrameric oligomeric states of the SpoIVFB: Pro-σ$^K$ complex. We believe that these oligomeric states are likely an artifact of detergent solubilization and weak lipid/detergent interactions between monomers in the purified protein preparations. It is worth noting that similar varieties of oligomeric states have been observed previously with detergent-solubilized SpoIVFB:Pro-σ$^K$ complexes using ion-mobility mass-

spectrometry, and the presence of higher order oligomers of SpoIVFB have been suggested from in vivo stepwise photobleaching experiments in intact sporulating B. subtilis[37]. Despite these previous reports of SpoIVFB oligomerization, it seems unlikely that the oligomeric states we observe here with cryo-EM represent functional oligomeric states that are sampled in sporulating B subtilis. Our MD simulation of monomeric SpoIVFB:Pro-σ$^K$ suggests that even a monomer of the complex is likely stable in a lipid bilayer environment. In an in vivo context it is important to consider that SpoIVFB is held inactive by interactions with additional inhibitory proteins SpoIVFA and BofA (Fig. 1; step 1). Further structural analysis is necessary to discern if the oligomeric states observed here are compatible with the quaternary structure of the larger SpoIVFA:BofA:SpoIVFB:Pro-σ$^K$ complex[13].

Finally, it is important to consider what conformational changes may occur throughout the process of Pro-σ$^K$ processing. A previous molecular modeling and B. subtilis sporulation-based study suggested that SpoIVFB can adopt closed and open conformations in vivo, and that this conformational shift can be manipulated by mutation of SpoIVFB residue F66[19]. This residue resides at the very apex of the membrane reentrant loop, and in our cryo-EM structure and MD simulations is situated directly beneath the acyl chains of LMNG/lipid (Fig. 6E). It was demonstrated that an F66A SpoIVFB variant could cleave Pro-σ$^K$ and allow for progression of sporulation even when the B. subtilis strain was devoid of the upstream proteases SpoIVB and CtpB[19]. These results suggest that during the process of relieving SpoIVFB inhibition and allowing processing of Pro-σ$^K$ (Fig. 1), SpoIVFB likely undergoes conformational changes to regulate entry of Pro-σ$^K$ into the enzyme active site. Further structural work with the SpoIVFA:BofA:SpoIVFB:Pro-σ$^K$ complex in various stages of activation is required to decipher how the activation of Pro-σ$^K$ processing proceeds, and what protein conformational changes mediate this process. This aspect of substrate engagement by SpoIVFB is distinct from that of distantly related S2P protease E. coli RseP, which requires ectodomain shedding from its substrate prior to active site entry and β-sheet augmentation[24,30]. Orthologs of RseP and SpoIVFB are broadly conserved among bacteria, including many pathogens in which their contributions to virulence make them potential therapeutic targets[3]. A better understanding of the mechanisms of substrate engagement by intramembrane proteases will facilitate translational efforts toward drug design.

## Methods

### Protein expression and purification

The genes encoding B. subtilis SpoIVFB-FLAG$_2$-His$_6$ and Pro-σ$^K$(1-127)-His$_6$ were placed behind individual IPTG-inducible T7 promoters in a custom-built pET expression vector[37]. BL21(DE3) E. coli were transformed with this expression plasmid and a single colony was inoculated into 100 mL Luria Broth (LB) medium with 50 μg/mL kanamycin and grown overnight at 37 °C. The following morning the starter culture was used to inoculate 6 L of LB medium with 50 μg/mL kanamycin and Antifoam-204 in baffled Fernbach shake flasks. The cultures were grown at 37 °C with agitation at 180 rpm to an OD$_{600}$ ~ 0.6 before reducing the temperature to 16 °C and inducing protein expression with 0.2 mM IPTG. Induced cells were allowed to continue growth at 16 °C overnight (~16 h) before the cultures were harvested by centrifugation. Bacterial pellets were stored at −80 °C until further use.

To purify SpoIVFB:Pro-σ$^K$ complexes the bacterial pellets were resuspended in lysis buffer containing 50 mM Tris (pH 8), 300 mM NaCl, 10% glycerol, 5 mM β-mercaptoethanol, 2 mM MgCl$_2$, 2 mM ATP, 1 μg/mL pepstatin A, 1 μg/mL leupeptin, 1 μg/mL aprotinin, 0.6 mM benzamidine, and 5000 units of Benzonase. Resuspended bacterial cells were lysed by sonication and large debris was removed by centrifugation at 4000 x g for 20 min before isolating membrane fractions by ultracentrifugation at ~100,000 x g for 1 h. Isolated membranes

were resuspended in lysis buffer and solubilized by stirring for 1 h at 4 °C with 1% lauryl maltose neopentyl glycol (LMNG). After detergent solubilization the insoluble material was removed by ultracentrifugation at 100,000 x $g$ for 1 h and the resulting supernatant was applied to Co$^{3+}$-Talon resin in a gravity fed column format. The resin was washed with ~10 column volumes of buffer A (50 mM Tris (pH 8), 300 mM NaCl, 10% glycerol, 5 mM $\beta$-mercaptoethanol, 2 mM MgCl$_2$, 2 mM ATP, 0.005% LMNG and 20 mM imidazole) before eluting bound proteins with buffer B (buffer A with 250 mM imidazole). The eluted proteins were concentrated in a 100 kDa MWCO spin concentrator and injected onto a Superdex 200 Increase 10/300 GL column equilibrated in 25 mM Tris (pH 8), 150 mM NaCl, 5 mM $\beta$-mercaptoethanol, 2 mM MgCl$_2$, 2 mM ATP and 0.005% LMNG. Peak fractions corresponding to the complex between SpoIVFB and Pro-σ$^K$ were collected and concentrated in a 100 kDa MWCO spin concentrator to ~8 mg/mL for cryo-EM analysis.

## Cryo-EM data collection and processing

Grids for cryo-EM imaging were prepared on a Vitrobot Mark IV by applying 3 μL of purified SpoIVFB:Pro-σ$^K$ at ~8 mg/mL to Quantifoil R2/2 200 mesh grids that had been glow-discharged for 45 s at 15 mA in a Pelco EasyGlow. Grids were blotted for 4–8 s at 4 °C, 100% humidity before being plunge frozen in liquid ethane cooled by liquid nitrogen. Data collection was performed on a Titan Krios with a K3 direct electron detector and Gatan BioQuantum GIF energy filter set to a 20 eV slit width at Purdue University (WT dataset) using Leginon[38], or at the University of Michigan (E44Q dataset) using SerialEM[39]. Movies were collected in counting mode with a pixel size of 0.872 Å and a total dose of 60 electrons/Å$^2$ (WT dataset) or a pixel size of 0.849 Å and a total dose of 50 electrons/Å$^2$ (E44Q dataset).

Movies were corrected for beam-induced motion by performing patch motion correction followed by patch CTF estimation in cryoSPARC[40]. Micrographs with CTF fit parameters worse than ~6 Å were discarded and further manual inspection was performed to remove obviously poor micrographs. Blob picking was used to generate initial 2D templates for template-based particle picking in cryoSPARC. Several rounds of 2D classification were performed to remove bad particles, and initial models for 3D classification were constructed using ab initio reconstruction in cryoSPARC. Where indicated the particles were transferred to RELION 4.0 for further 3D classification and signal subtraction[41], before finally being reimported to cryoSPARC for a final round of non-uniform refinement[42]. Resolutions of all maps were calculated using the gold-standard FSC and local resolutions were calculated in cryoSPARC.

AlphaFold2-Multimer models[43] of SpoIVFB and Pro-σ$^K$ were initially rigid-body docked into the WT cryo-EM map using UCSF Chimera[44]. The model was manually adjusted in COOT[45] to properly fit the cryo-EM density and refined in real-space using phenix.real_space_refine in the PHENIX software suite[46]. Iterative rounds of real-space refinement in PHENIX and manual model adjustment in COOT were performed to optimize the overall model geometry and fit to the experimental cryo-EM map. A comparison of the final cryo-EM derived atomic model and the initial AlphaFold model reveals an overall RMSD of 2.6 Å between SpoIVFB:SpoIVFB and an RMSD of 7.3 Å between Pro-σ$^K$:Pro-σ$^K$. The largest difference between the experimental cryo-EM structure and the AlphaFold prediction of SpoIVFB:Pro-σ$^K$ lies in the configuration and positioning of the Pro-σ$^K$ pro-sequence (alpha-helical in AlphaFold prediction, and beta-sheet configuration in the cryo-EM derived model). To refine the model of E44Q SpoIVFB:Pro-σ$^K$ we rigid body docked the WT atomic model into the experimental density map, and iteratively refined the model with manual adjustment in COOT and real-space refinement in PHENIX. In both the WT and E44Q maps of SpoIVFB:Pro-σ$^K$ the regions encompassing the C-terminus of Pro-σ$^K$ (residues 119–126) and a loop in the SpoIVFB CBS domain (residues 253–263) were of

particularly lower resolution than the rest of the complex (Suppl. Figs. S3D and S6D). Sidechains of residues in these regions were trimmed back to the C$_\beta$ carbon. All models were assessed for appropriate stereochemical properties and fit to the experimental cryo-EM map using MolProbity[47] as implemented in PHENIX. Figures were created using UCSF Chimera, UCSF ChimeraX[48], and Blender 2.8.

## Site-directed mutagenesis and Pro-σ$^K$ cleavage assay

Site-directed mutagenesis of SpoIVFB and Pro-σ$^K$ was performed with a Q5 mutagenesis kit from NEB using standard manufacturer protocols (oligonucleotide primer sequences used in this study are included in the accompanying Source Data file). Plasmids encoding variants of SpoIVFB or Pro-σ$^K$ were verified through Sanger sequencing at the RTSF Genomics Core at Michigan State University. For Pro-σ$^K$ cleavage assays in *E. coli* BL21(DE3), cells were transformed with an expression vector containing WT or variant SpoIVFB and Pro-σ$^K$ (described above in protein expression) and a single colony was grown overnight in 4 mL of LB medium with 50 μg/mL kanamycin at 37 °C. The following morning 0.5 mL of the overnight culture was diluted into 30 mL of LB medium with 50 μg/mL kanamycin and the culture was grown with shaking at 37 °C. Once the cultures reached an OD$_{600}$ of ~0.8, IPTG was added to a final concentration of 1 mM and growth was continued for 2 h before harvesting 1 OD$_{600}$ of cells by centrifugation at 3500 x $g$ for 5 min.

To prepare cell lysates for Western blot analysis the pellets from 1 OD$_{600}$ of culture were resuspended in 200 μL of lysis buffer (50 mM Tris (pH 8), 5 mM MgCl$_2$, 1 mM EDTA, 0.5 mg/mL lysozyme, 1 μg/mL pepstatin A, 1 μg/mL leupeptin, 1 μg/mL aprotinin, 0.6 mM benzamidine, and 250 units of benzonase). Resuspended pellets were incubated at 37 °C for 10 min before mixing 45 μL of resuspended cells with 9 μL of 6X SDS-PAGE Laemmli buffer containing DTT. The samples were then boiled for 3 min before electrophoresis on a 17% reducing SDS-PAGE gel and transfer to a nitrocellulose membrane in a wet-transfer apparatus according to manufacturer protocols (BioRad). Western blots were blocked and then probed with an HRP-conjugated anti-pentaHIS antibody according to manufacturer protocols (Qiagen), and chemiluminescence detection was performed with an ECL Prime detection kit (Amersham) on an Azure Sapphire Biomolecular Imager. Western blot band intensity was analyzed in ImageJ[49]. Experiments were performed in triplicate starting from initial plasmid transformations into *E. coli*, and statistical analyses including error bar calculation for bar graphs and two-tailed unpaired t-tests were performed in GraphPad Prism 9.

## In vivo photocrosslinking

To perform photocrosslinking in intact *E. coli*, amber codons (TAG) were introduced into SpoIVFB or Pro-σ$^K$ using the site-directed mutagenesis protocol outlined above. The unnatural amino acid p-azidophenylalanine (pAzF) was incorporated at these amber positions by co-transforming the SpoIVFB:Pro-σ$^K$ expression plasmid and pEVOL-pAzF[50] into BL21(DE3) *E. coli* and plating onto LB agar plates containing 25 μg/mL chloramphenicol and 50 μg/mL kanamycin. A single colony was grown overnight in 30 mL LB media with the same antibiotics at 37 °C. The following morning 300 μL of the overnight culture was diluted into 30 mL of LB media with the same antibiotics and grown at 37 °C to an OD$_{600}$ ~ 0.5, at which time 300 μL of 100 mM p-AzF dissolved in 300 mM NaOH was added to the culture and growth was continued for another 30 min to allow uptake of the unnatural amino acid. Then the culture was adjusted to a final concentration of 0.1% arabinose and 1 mM IPTG and growth was continued for an additional 2 h, after which spectinomycin was added to a final concentration of 100 μg/mL to stop protein translation. Bacterial culture corresponding to 1 OD$_{600}$ was pelleted by centrifugation at 4000 x $g$ for 5 min and resuspended in ice-cold PBS and

dispensed into 24 well tissue culture plates on ice. The resuspended bacteria were then irradiated with 254 nm UV light for 5 min before centrifugation at 4000 x $g$ for 5 min and flash freezing in liquid nitrogen. Bacterial pellets were then processed and analyzed by Western blot using the same procedure described above for Pro-$\sigma^K$ cleavage assays.

## Disulfide crosslinking

Disulfide crosslinking between site-specifically introduced cysteine residues was performed following a previously published protocol with minor modifications[51]. Briefly, BL21(DE3) *E. coli* transformed with plasmids containing single-cysteine variants of both SpoIVFB and Pro-$\sigma^K$ were grown in LB media and protein expression induced with IPTG. After 2 h of expression at 37 °C equivalent amounts of cells were pelleted, resuspended in LB media and mixed with chloramphenicol (200 µg/ml) and 1,10-phenanthroline (3 mM), collected by centrifugation and washed with 10 mM Tris-HCl pH 8.1 containing 3 mM 1,10-phenanthroline, and suspended in 10 mM Tris-HCl pH 8.1. A control experiment titrating with $Cu^{2+}$(phenanthroline)$_3$ demonstrated that a concentration of 1 µM oxidizing reagent generated significant crosslinking in cysteine pairs designed as positive controls, and minimal or no crosslinking with a cysteine pair designed as a negative control (Suppl. Fig. S8B). Samples were treated with 1 µM $Cu^{2+}$(phenanthroline)$_3$ or 3 mM 1,10-phenanthroline (as a control) for 5 min at 37 °C, followed by incubation with neocuproine (12.5 mM) for 5 min at 37 °C. Cells were lysed and proteins were precipitated by the addition of trichloroacetic acid (5%) on ice. Precipitate proteins were sedimented by centrifugation and the pellet was washed with ice-cold acetone before being pelleted again by centrifugation. Precipitated protein pellets were dried for 5 min at 25 °C and resuspended in buffer (100 mM Tris-HCl pH 7.5, 1.5% SDS, 5 mM EDTA, 25 mM *N*-ethylmaleimide) for 30 min at 25 °C. Resuspended proteins were mixed with an equal volume of sample buffer (25 mM Tris-HCl pH 6.8, 2% SDS, 10% glycerol, and 0.015% bromophenol blue) with (control sample) or without ($Cu^{2+}$(phenanthroline)$_3$-treated sample) 100 mM DTT and incubated at 98 °C for 5 min prior to immunoblot analysis. The entire procedure outlined above was also completed with SpoIVFB lacking a His$_6$ tag to demonstrate His$_6$-tagged Pro-$\sigma^K$ in a crosslinked species (Suppl. Fig. S8C) and with single-cysteine variants of either SpoIVFB or Pro-$\sigma^K$ as negative controls (i.e., the other protein co-expressed, but cysteine-less), all of which showed no disulfide crosslinking (Suppl. Fig. S8D, E).

## Molecular dynamics

To perform all-atom MD simulations a monomer of SpoIVFB:Pro-$\sigma^K$ from the WT cryo-EM structure was embedded into a lipid bilayer composed of a lipid mixture containing 53% POPE, 30% DAG (16:0, 18:1), 14% POPG, and 3% cardiolipin (16:0, a15:0, 16:0, a15:0) to resemble the distribution of lipids previously found in *B. subtilis* membranes[29], and the systems were solvated using a TIP3P water model, and charge neutralized with ~150 mM KCl using CHARMM-GUI[52]. Four independent systems were assembled in this fashion using separate independent runs of CHARMM-GUI. The final systems containing ~230,000 atoms each were energy minimized using standard output scripts from CHARMM-GUI at 310 K and 1 atm under an NPT ensemble in OpenMM[53] using the Charmm36m forcefield[54]. The final unrestrained 250 ns production runs were similarly performed in OpenMM with the Charmm36m forcefield using the Langevin integrator[55] and Monte Carlo membrane barostat[56]. The force-based switching method was used for smoothing vdW interactions over the distance range 10-12 Å, and the particle-mesh Ewald (PME) method was used for long-range electrostatic interactions[57]. The simulation used periodic boundary conditions and a timestep of 2 femtoseconds. Snapshots from the trajectory were saved every 2 picoseconds and analyzed with VMD[58].

## Reporting summary

Further information on research design is available in the Nature Portfolio Reporting Summary linked to this article.

## Data availability

Data supporting the studies presented here are available upon request from the corresponding author. Atomic coordinates for the two structures reported in this publication have been deposited in the Protein Data Bank (PDB) under accession codes 8VJL (WT SpoIVFB:Pro-$\sigma^K$); 8VJM (E44Q SpoIVFB:Pro-$\sigma^K$). Cryo-EM maps have been deposited in the Electron Microscopy Data Bank (EMDB) under accession codes EMD-43288 (WT SpoIVFB:Pro-$\sigma^K$); EMD-43289 (E44Q SpoIVFB:Pro-$\sigma^K$). The AlphaFold model of SpoIVFB and Pro-$\sigma^K$ used as an initial template for model building is available at the AlphaFold Protein Structure Database [SpoIVFB: https://alphafold.ebi.ac.uk/entry/P26937] [Pro-$\sigma^K$: https://alphafold.ebi.ac.uk/entry/P12254]. Files relevant for visualizing and reproducing the molecular dynamics simulations have been uploaded to and are available on Zenodo. Replicate 1: [https://doi.org/10.5281/zenodo.11479025], Replicate 2: [https://doi.org/10.5281/zenodo.11479394], Replicate 3: [https://doi.org/10.5281/zenodo.11479422], Replicate 4: [https://doi.org/10.5281/zenodo.11479460], Simulation with "pore water": [https://doi.org/10.5281/zenodo.11479490], Simulation in POPE/POPG mixture: [https://doi.org/10.5281/zenodo.11479578. Source data are provided with this paper.

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

## Acknowledgements

Research reported in this publication was supported by the National Institute of General Medical Sciences of the National Institutes of Health under Award Number R35GM146721 to BJO and R01GM043585 to LK. The content is solely the responsibility of the authors and does not necessarily represent the official views of the National Institutes of Health. Electron Microscope support at Purdue University was established under the NIH Midwest Cryo-EM Consortium grant number 1U24GM116789-01A. We would like to thank Dr. Sundharraman Subramanian for help with electron microscope operation at the RTSF Cryo-Electron Microscopy Facility at Michigan State University. We are also grateful for the NIH Midwest Cryo-EM Consortium and Dr. Thomas Klose and Dr. Wen Jiang for providing microscope time and assistance with data collection at Purdue University, as well as Dr. Alexandrea Rizo, Dr. Ashleigh Raczkowski, and Dr. Vinson Lam for assistance with cryo-EM data collection at the University of Michigan Life Sciences Institute.

## Author contributions

M.A.O. and H.J.T.P. performed site-directed mutagenesis, in vivo cleavage assays and statistical analysis. S.M. performed site-directed mutagenesis, disulfide crosslinking and immunoblot analysis. B.J.O. performed protein purification, cryo-EM imaging and data analysis, in vivo photocrosslinking, and molecular dynamics simulations. B.J.O. and L.K. conceived the project and wrote the manuscript.

## Competing interests

The authors declare no competing interests.
