## [Peer Review File · Nature Communications]

Substrate Engagement by the Intramembrane
Metalloprotease SpoIVFBREVIEWER COMMENTS

Reviewer #1 (Remarks to the Author):

The manuscript from the Orlando group "Substrate Engagement by the Intramembrane Metalloprotease SpoIVFB" is a beautiful piece of work and the culmination of in vivo and in vitro experiments from the Kroos, Rudner, Cutting, and Losick labs dating back a quarter century: The discovery of the pro-domain on SigK in 1989, the discovery that SpoIVFB is likely to be a membrane-embedded metalloprotease in 1999, and the in vitro reconstitution of SpoIVFB-mediated Pro-sigK processing in 2009.

The manuscript is well written and balanced in the interpretations that can be drawn from the structure. The structure is nicely validated by many previous observations from in vivo studies: the location of the Pro-sigK cleavage site adjacent to the catalytic residues in SpoIVFB; the importance of the interdomain linker of SpoIVFB in binding Pro-sigK; and the minor role of the CBS domain. The authors further validate the structure with alanine- and tryptophan-scanning mutagenesis combined with pro-sigK cleavage assays in vivo. In addition, they perform disulfide- and photo-crosslinking experiments that further validate their structure.

This is a terrific study and I eagerly await the structure of SpoIVFB in complex with its regulators SpoIVFA and BofA.

I have several minor comments that the authors need only consider (i.e. they can make changes as they see fit).

1) Line 15: change "bound to the proteolytic substrate Pro-sigK" to "bound to its native substrate Pro-sigK"

2) Line 55: Ref 28 could be inserted at the end of this sentence. The experiment in the Guadiana paper where Pro-sigK is pulled down with SpoIVFB only in the presence of SpoIVB is a striking result that fits here. I had forgotten about this experiment!

3) Line 60: Consider "an elegant developmental checkpoint"

4) Lines 83-85 and first paragraph of the Discussion: The authors emphasize the role of beta-sheet augmentation as a conserved mechanism of substrate engagement for membrane-embedded proteases. I think this mechanism merits even more emphasis - this is a strong feature of the paper. I recommend a supplemental figure comparing substrate engagement for SpoIVFB/Pro-sigK, gamma-secretase/Notch, and Rhomboid(GlpG)/inhibitor structures. I will certainly show such a figure in the class that I teach on membrane-embedded proteases.

6) Line 151: should this be antiparallel beta-strands?

7) Line 224: H209 (should this be F209?)

8) Line 230: H209A (should this be F209A?)
OR Is the figure incorrect?

9) Line 341-2: The concluding sentence about the CBS domain strikes me as too strong given the in vivo data showing its minor role in signaling and SigK activation. Consider changing to: "The contribution of the CBS domain to pro-sigK processing in the context of the full SpoIVFA:BofA:SpoIVFB complex will require further investigation."

10) The discussion of the SpoIVFB:Pro-sigK tetramer and the potential that these are artifacts of detergent solubilization and weak lipid/detergent interactions is both balanced and diplomatic. The

photobleaching experiments referenced are difficult to interpret as this assay is very challenging in bacteria.

A quick AlphaFold-multimer with SpoIVFB-SpoIVFA-BofA suggests that the two regulators of SpoIVFB would likely partially occlude the oligomeric interfaces observed in the cryo structure.

11) I hope the stability of SpoIVFB will be better behaved in the presence of SpoIVFA and BofA.

12) The discussion might be strengthened by a few sentences about the RseP subfamily of site-2 proteases. These are significantly more broadly conserved among bacteria and play a key role in stress-response signal transduction pathways in gram-negative and -positive pathogens. The authors might emphasize that the regulated step in intramembrane proteolysis of anti-sigma factor substrates is ectodomain shedding. But specificity is likely imparted by beta-sheet augmentation without steric clashes. As such, RseP/RasP proteases are constitutive proteases but have specific substrate requirements.

Reviewer #2 (Remarks to the Author):

In this study, Orlando et al report the cryoEM structure of the S2P SpoIVFB from *B. subtilis* in complex with its substrate pro- σ^K as well as follow up biochemical experiments and simulations to support the experimental model and implicated mechanism of substrate recruitment and catalysis. This is a major leap for our understanding of S2Ps and intramembrane proteases more generally. Namely, all three intramembrane protease classes seem to use hybrid β -sheet formation between the substrate and enzyme as a driving force for luring the membrane-bound substrate into their hydrophilic active sites. The manuscript is generally well explained and thorough. The crosslinking (photo- and cys-) experiments have flaws that compromise interpretation, however. Some of the figures should be amended to convey more readily what is described in the narrative of the text. Finally, queries/curiosities for possible inclusion in a revised manuscript are listed below.

Structure

1. The statement that mjS2P and the SpoIVFB are similar yet have an rmsd of 3.5 Å is a contradiction. The overall topology is similar but extensive differences, e.g. in the pitch of TM3, TM6, plus a helix on the side is new, are apparent. Is the structural difference an artifact of mjS2P having been truncated or could there be a conformational change between apo and pro- σ^K bound? (see also question about the simulations below) Perhaps more could be added to the manuscript about this.
2. Please include a zoom-in of the active site w/stick views of key residues so that protease aficionados can more easily deduce the contributors to the implied hydrolytic chemistry. Right now the most detail of the active site is Fig 4B. (Seems like the LMNG density can be omitted and the active site region be zoomed in).
3. Figure 5A is challenging to interpret. Perhaps as surface representation of the different sections and/or electrostatic maps, combined with judicious use of stick for residues would help with this.
4. Regarding methods, please include the rmsd of the final experimental models with the initial AF2 models used as initial models (and maybe rmsd of AF2 model and mjS2P as well, if pertinent)? As it becomes more common practice to use AF2 as a starting model for an experimental data set, I think this information is interesting to mention in manuscripts (was it mainly the side chains that needed adjusting or was more backbone rebuilding required to match density). Obviously, this is most interesting in cases where the starting and ending model differ notably, but either way this is a good data point since going forward AF2 models for S2P orthologs will be updated as a result of this new structure and 'historical' AF2 structures will not be easily compared.

Biochemistry/Simulations

1. It's remarkable that many point mutations have no effect on activity, but in the case where enzyme

activity was abrogated e.g. Y204A/R208A, it would seem prudent to include blots showing that the mutants were actually expressed and that loss of activity observed is not due to enzyme misfolding and degradation.

2. None of the photo- or Cys-crosslinking blots have a ladder or another way to show that the crosslinked species are the two proteins of interest and not multimers of one component. This is a major flaw of the paper.
3. Full blots for both photo- and Cys-crosslinking blots should also be presented in the supplemental data, presumably this is required by the journal.
4. Please move Supp Fig. S8B to the main Figure 5 and amend to highlight where the strongest crosslinks are found in the structure, e.g. Y215/N53. Otherwise, the text is hard to follow.
5. I will leave the quality of the simulations to an expert reviewer in that methodology, but interpreting the simulations as presented is not straightforward with the images provided in Fig. 6A. Perhaps an electrostatic surface representation would complement the apparent trajectory of the waters into the active site. Further zoom into the active site with the substrate bound alongside the waters (and labeling all residues even after 90 ° rotation please) might assist with interpretation.
6. Do the authors think water binds first and the substrate displaces waters already present? Is this an additional driving force (entropy/water release) that contributes to substrate binding in the active site?

Other Comments

1. I am not concerned about the structure being of an apo SpoIVFB with pro- σ K. Surely the authors have tried spiking with Zn at the time of induction. If the hypothesis is right, namely, Zn is labile and lost during purification, presumably you would get more cleaved pro- σ K and less intact complex if you spiked with Zn. A comment along these lines could mitigate the caveat about the study early in the paper.
2. Based on the structure, do the authors have any insight into S2Ps that contain other domains, e.g. PDZ? It might be nice to broaden the scope of the findings for SpoIVFB to others S2Ps in the discussion.
3. One of the interesting findings specific to S2Ps is that the interdomain linker plays a major role in mediating SpoIVFB -pro- σ K interactions. How conserved is this region, say, across bacterial S2Ps with a similar overall domain architecture to SpoIVFB? If conservation is low, then there might be different ways different types of S2Ps engage their substrates, which would be worth mentioning as a limitation of the present study but worthwhile for future follow up.
4. As alluded to in several places above, the finding that three classes of IPs--SpoIVFB, RseP+Rhomboid, and γ -secretase—engage substrates by forming a hybrid β -strand between substrate and enzyme is one of the most powerful broader implications of this paper. One of the fundamental questions surrounding IPs has been how a transmembrane/mostly hydrophobic helix would find its way into a chemically-incompatible hydrophilic active site for cleavage, and now it is apparent! This commonality is mentioned in the first paragraph of the discussion but I think the paper would benefit from further expansion. To what extent might the driving force for helix unwinding be to form a hybrid β -strand/enthalpic vs displacing waters/entropic? If all IPs use this 'mechanism' then why are some IPs processive and others specific?

Reviewer #3 (Remarks to the Author):

M. A. Orlando and colleagues have determined the high-resolution structure of SpoIVFB in complex with its proteolytic substrate Pro- σ K using cryo-electron microscopy single particle analysis. SpoIVFB is a member of the S2P family of intramembrane metalloproteases, characterized by a C-terminal CBS domain. However, in the existing resolved structures of other S2P family intramembrane metalloproteases, the CBS domain has consistently been omitted for simplicity and has never been resolved within a complex.

This study presents a detailed molecular view of substrate engagement and positioning within an S2P

intramembrane metalloprotease, offering insights into the mechanism of Pro- σ K cleavage facilitated by SpoIVFB. The authors further enriched these findings with a comprehensive mutational analysis and protein cross-linking experiments to elucidate the protein-protein interactions essential for the precise positioning of Pro- σ K during intramembrane proteolysis.

Additionally, they employed molecular dynamics simulations to assess the stability of the structure within a model membrane, revealing the formation of a water-filled cavity.

The manuscript is well-written, featuring clear, beautifully presented, and straightforward to comprehend figures. I thoroughly enjoyed engaging with this high-quality work, and it merits consideration for publication in Nature Communications, given its novelty and the fascinating insights it provides.

Nonetheless, the section on molecular dynamics simulations should be improved. The authors have conducted a single 250 ns molecular dynamics simulation to visually demonstrate the influx of water into a cavity and the potential binding site for a PE lipid. This is a limited approach to exploring these complex interactions.

Regarding the simulations, I appreciated the author's effort to include them. I believe the depiction of a water channel adds a compelling element to the narrative, suggesting a plausible mechanism by which water facilitates the hydrolytic cleavage of the Pro- σ K peptide backbone. However, the presentation is limited to snapshots from the MD simulations rather than a comprehensive data representation. Moreover, the study relies on a single MD simulation with an unsuitable membrane model composition. I would advise the author to conduct at least three independent simulations (in this case building three times from CHARMM-GUI) and to calculate average properties such as hydration numbers (the number of water molecules over time within the cavity) and contact occupancy between water molecules and residues. Additionally, analyzing the interaction energy would provide a detailed characterization of average properties and the hydrophilic route.

Furthermore, I have concerns regarding the choice of model membrane utilized in this study. The authors opted for a 3:1 mixture of POPE:POPG. It is known that PE is not a bilayer-forming lipid, and at concentrations greater than 60%, vesicles are not stable in experiments due to induced defects and phase transitions. Additionally, I question the rationale behind using PG (e.g., I would have chosen POPC instead) or of this specific composition in general. Is there a specific reason for its inclusion? How did the authors arrive at this particular membrane composition?

For these reasons, I recommend that the authors explore alternative membrane compositions and if possible, more than one. The protein may interact differently with different membrane environment, potentially impacting protein orientation and the water channel formation. For each membrane composition tested, a minimum of three molecular dynamics simulations should be conducted to ensure robust and statistically reliable results.

The authors' assertion that the protein traps a POPE lipid molecule lacks statistical evidence since it is based solely on a single MD simulation where the lipid immediately migrates to a specific site. To substantiate this claim, the authors should conduct multiple independent simulations where the lipid is initially positioned away from the pocket. For instance, after the lipid moves to the pocket, the author could remove it and see if another lipid returns in, offering a more rigorous test of its affinity. Additionally, I expected the authors to delve into characterizing the nature of the lipid-protein interaction—whether it is dependent on the lipid's tail or head or merely driven by hydrophobic interactions. This could be achieved through detailed analyses such as contact occupancy and interaction energy, providing deeper insight into the interaction dynamics.

Furthermore, the methods section lacks references for the CHARMM36 force field, thermostat, barostat, and algorithms for calculating electrostatic interactions. Additionally, the water model used in the simulations is not specified. Furthermore, data about the molecular dynamics simulations are not accessible for reproducing or visualizing the trajectory. I recommend making at least the minimum required files for reproducing the simulations publicly available. Numerous platforms, such as Zenodo.org, can provide a Digital Object Identifier (DOI) for this purpose.

Reviewer #4 (Remarks to the Author):

The manuscript entitled "Substrate Engagement by the Intramembrane 1 Metalloprotease SpoIVFB" by Orlando et al. mainly focuses on cryo-EM-based structural characterization of Metalloprotease SpoIVFB bound to the substrate Pro- σ K from *Bacillus subtilis*, which is one of first cryo-EM structure of SpoIVFB bound Pro- σ K. However, this structure was characterized with full-length Pro- σ K. From these structural studies, it is not clear how enzymatically active SpoIVFB was able to bind with full-length Pro- σ K, which is a substrate of SpoIVFB. The authors also performed cryo-EM structure calculation of mutated SpoIVFB. Additionally, mutational and protein crosslinking analyses pointed to several key amino acid residues of protein-protein interactions between SpoIVFB and Pro- σ K.

The authors performed a good amount of work in this manuscript, which covers protein expression & purification, mutation, crosslinking studies, and cryo-EM to calculate the 3D structure. Authors resolved high-resolution structures of Wt and mutated SpoIVFB with Pro- σ K. This manuscript was well-written, and overall, the representation was good. However, more clarifications are required during the data represented.

There are some flaws, which are described below.

The major drawback of this work was that Pro- σ K is a substrate of SpoIVFB, which should be proteolytically cleaved by SpoIVFB. However, the authors resolved the full-length structures of SpoIVFB with Pro- σ K. This raises an important question: Did the authors select inactive SpoIVFB for structure calculation? If authors were aware of this, why did they not take suitable steps to purify the active protein? How is this structural information from inactive proteins relevant? The authors performed photo-crosslinking and disulfide crosslinking. Samples were used to perform immunoblot analysis; It may provide positive results due to disulfide crosslinking. The author should clarify how crosslinking affects their immunoblot and other biochemical analyses. I hope the authors did not use this crosslinked protein for their cryo-EM analysis. One possibility is that the authors resolved the full-length Pro- σ K with SpoIVFB due to crosslinking.

I am seriously concerned about cryo-EM data processing. Author mentioned in Supplemental Figure S2A&B that authors observed several oligomeric states of Pro- σ K with SpoIVFB; oligomeric states, including dimers, trimers, and tetramers of SpoIVFB:Pro- σ K (Suppl. Fig. S2A&B) were observed in 2D class averages. How did they identify dimers from tetramers and trimers? Generally, from the top view, it is easy to identify tetramer and trimer, but I am unable to see any dimeric top views. However, isolating dimers from trimers and tetramers from the side views of 2D averages is challenging. How did the authors isolate side views of this complex? It will be better to run 2D averages of the individual 3D model data sets and show that particles were segregated based on dimer, trimer, and tetramer.

How did the authors generate the initial model? In Figure S2C, they pointed out that "attempts to reconstruct the SpoIVFB:Pro- σ K trimers were unsuccessful, likely due to severe orientation bias of this particular species". Could they provide the number of particles and angular distribution of the trimeric state?

Figure 2d, 2nd, and last class average showed only one extra density from the membrane part. I can expect this 2d came from the final 3d structure, and a small extra density should be for the non-member part of SpoIV and Pro- σ K, but suppl Fig 2b, the last class avg (blue circle) mentioned it is dimer... this is contradicting author's claim.

The authors showed in Figure 3 and Supplemental Figure S7 and Figure 4 the structure of the SpoIVFB intramembrane protease and the zinc identified in the mjs2P structure. The binding of the pro-sequence of Pro- σ K to SpoIVFB showed LMNG detergent. However, this 3D structure should be surrounded by LNMG. Why did the authors observe only one LNMG density? Authors should show the atomic model fitted map and point to the Zn density as well as LNMG density. It may have an important role if this LNMG density is located inside the TM domain. Did the authors perform any experiments, or did they have any explanations for the role of LNMG and lipids inside the TM? Is there any evidence that LNMG or lipid is required for activation?

The authors proposed, "In total, the physical interaction between monomers in the WT SpoIVFB:Pro- σ K tetramer is quite limited ($\sim 488\text{\AA}^2$ for interface 1, $\sim 510\text{\AA}^2$ for interface 2), which makes for fragile interactions between SpoIVFB:Pro- σ K monomers that are largely mediated by lipid/detergent....." However, the authors did not perform any proper experiments to support their claim.

The authors performed two reconstructions of WT and E44Q SpoIVFB: Pro- σ K complexes and compared the final cryo-EM structures of WT and E44Q SpoIVFB:Pro- σ K complexes and claimed that E44Q SpoIVFB:Pro- σ K is approximately one-half the size of the tetrameric WT complex. However, there are no proper figures where the reader can see the clear difference between WT and E44Q SpoIVFB:Pro- σ K complexes. A proper figure should be included in the manuscript.

The authors focused on different oligomeric states of WT and E44Q SpoIVFB:Pro- σ K in the entire manuscript. However, no proper biophysical experiments were performed to validate their claim. It will be better to perform SEC-MALS or other biophysical experiments to support their claim. It will also be better to perform 2D class averages of dimeric or trimeric 3D structures and support their claim. More severe problem, the authors didn't mention how one single mutation, E44Q, could abolish the formation of trimers and tetramers.

Basically, the non-membrane part of the SpoIVFB:Pro- σ K complex looks higher resolution than the TM region. Authors should perform local resolution calculations. RMSD $\sim 3.5\text{\AA}$ is a bit higher side. Several helix and loop regions of the non-membrane part of the SpoIVFB:Pro- σ K complex of both maps are not fitted properly. Even at the high threshold, the helix and loop regions of the atomic model are outside of the EM density map (see the attachment). I am concerned about the model building. When EM densities are not visible, how accurate is an atomic model? Also, at higher thresholds, many extra densities are observed. In these circumstances, how confident were the authors about LNMG densities? The author should show the map-to-model fitting, angular distribution, and FSC with and without a mask.

The authors claimed that H43, H47, and D137 of SpoIVFB are conserved with those of mjs2P; the cryo-EM map of WT SpoIVFB:Pro- σ K does not show any appreciable density for an ion being coordinated by these residues (Fig. 3B). However, in figure 3B represented Zn is binding with H43, H47, and D137. This is very misleading. The authors also claimed that they were able to co-purify WT SpoIVFB with full-length unprocessed Pro- σ K despite displaying robust proteolytic activity in *E. coli*. Also, extra Zn ions do not activate the proteolytic activity. How does this complex, if physiologically relevant? The author should perform an enzyme activity assay to demonstrate that the entire complex is enzymatically active.

Based on the previous report and their experiment, the mentioned "H206A and H209A variants processed Pro- σ K comparable to WT SpoIVFB". However, it is unclear what they wanted to conclude from this experiment.

The cryo-EM data and MD simulation data showed that the membrane plays a role in stabilizing the complex. What type of lipid can play a role in enzyme stabilization? The authors can elaborate on that.

Additionally, authors' biochemical data showed protein is inactive in the presence of detergent. How do authors justify lipid can play a role in enzyme stabilization and activation.

Wt

Wt

Volume Viewer

Features Data Tools

local_filter_map.mrc #0 320³ step 1

je -0.57 -0.04 Level 0.161 Color

Intensity 1

Transparency 0.30

surface mesh solid

Center Orient Close Help

Remove Dust

local_filter_map.mrc (#0)

Options Close Help

Mutant

Mutant

Volume Viewer

Features Data Tools

sharpened_map.mrc #0 320³ step 1

-0.597 - 0.92 Level 0.201 Color

ess 1

parency 0.36

surface mesh solid

Center Orient Close Help

Dear *Nature Communications* Editorial Staff,

Thank you for facilitating the review of our manuscript titled "Substrate Engagement by the Intramembrane Metalloprotease SpoIVFB". We are very pleased that three reviewers had overall positive comments. After analyzing the reviewer's comments and making the suggested corrections, we believe that our manuscript has been significantly strengthened. Highlighted below are point-by-point responses to the critiques raised by each reviewer. The original comments and suggestions from the reviewers are highlighted in *italicized and indented text*. Our responses and any corrective actions that have been taken with the manuscript are **highlighted in bold beneath each reviewer critique**.

Reviewer #1 (Remarks to the Author):

The manuscript from the Orlando group "Substrate Engagement by the Intramembrane Metalloprotease SpoIVFB" is a beautiful piece of work and the culmination of in vivo and in vitro experiments from the Kroos, Rudner, Cutting, and Losick labs dating back a quarter century: The discovery of the pro-domain on SigK in 1989, the discovery that SpoIVFB is likely to be a membrane-embedded metalloprotease in 1999, and the in vitro reconstitution of SpoIVFB-mediated Pro-sigK processing in 2009.

The manuscript is well written and balanced in the interpretations that can be drawn from the structure. The structure is nicely validated by many previous observations from in vivo studies: the location of the Pro-sigK cleavage site adjacent to the catalytic residues in SpoIVFB; the importance of the interdomain linker of SpoIVFB in binding Pro-sigK; and the minor role of the CBS domain. The authors further validate the structure with alanine- and tryptophan-scanning mutagenesis combined with pro-sigK cleavage assays in vivo. In addition, they perform disulfide- and photo-crosslinking experiments that further validate their structure.

This is a terrific study and I eagerly await the structure of SpoIVFB in complex with its regulators SpoIVFA and BofA.

We thank the reviewer for their gracious comments. Indeed, various experiments from the Kroos, Rudner, Cutting, Losick, and other labs over the past 25+ years have provided unprecedented mechanistic insight into the unusual SpoIVFB intramembrane protease. However, a general lack of structural data on this complex has hampered development of deeper mechanistic insights for some time. We are glad that reviewer #1 agrees that these highly sought-after structures of SpoIVFB shed light on critical areas that lacked mechanistic clarity (roles of the interdomain linker, CBS domain, etc...), and that the functional data presented further complements and validates the structural data.

I have several minor comments that the authors need only consider (i.e. they can make changes as they see fit).

1) Line 15: change "bound to the proteolytic substrate Pro-sigK" to "bound to its native substrate Pro-sigK"

The change has been made.

2) Line 55: Ref 28 could be inserted at the end of this sentence. The experiment in the Guadiana paper where Pro-sigK is pulled down with SpoIVFB only in the presence of SpoIVB is a striking result that fits here. I had forgotten about this experiment!

A reference to Ramírez-Guadiana, F.H. et al. (2018) has been added at the end of the sentence.

3) Line 60: Consider "an elegant developmental checkpoint"

The change has been made.

4) Lines 83-85 and first paragraph of the Discussion: The authors emphasize the role of beta-sheet augmentation as a conserved mechanism of substrate engagement for membrane-embedded proteases. I think this mechanism merits even more emphasis - this is a strong feature of the paper. I recommend a supplemental figure comparing substrate engagement for SpoIVFB/Pro-sigK, gamma-secretase/Notch, and Rhomboid(GlpG)/inhibitor structures. I will certainly show such a figure in the class that I teach on membrane-embedded proteases.

It appears that beta-sheet augmentation is a likely conserved mechanism of substrate engagement across membrane embedded (and soluble) proteases. As requested, we have generated an additional figure (Supplemental Figure S10) comparing the beta-sheet augmentation mechanism of substrate/ligand capture by SpoIVFB, RseP, GlpG, and γ -secretase. Attention to this new supplemental figure has been pointed out throughout the first paragraph of the discussion, highlighting the similarities of substrate engagement across broad classes of intramembrane proteases.

6) Line 151: should this be antiparallel beta-strands?

Yes, the change has been made.

7) Line 224: H209 (should this be F209?)

See below...

8) Line 230: H209A (should this be F209A?) OR Is the figure incorrect?

Thank you for catching the errors above. All instances are now correctly referred to as F209.

9) Line 341-2: The concluding sentence about the CBS domain strikes me as too strong given the in vivo data showing its minor role in signaling and SigK activation. Consider changing to: "The contribution of the CBS domain to pro-sigK processing in the context of the full SpoIVFA:BofA:SpoIVFB complex will require further investigation."

The requested change has been made.

10) The discussion of the SpoIVFB:Pro-sigK tetramer and the potential that these are artifacts of detergent solubilization and weak lipid/detergent interactions is both balanced and diplomatic. The photobleaching experiments referenced are difficult to interpret as this assay is very challenging in bacteria. A quick AlphaFold-multimer with SpoIVFB-SpoIVFA-BofA suggests that the two regulators of SpoIVFB would likely partially occlude the oligomeric interfaces observed in the cryo structure. I hope the stability of SpoIVFB will be better behaved in the presence of SpoIVFA and BofA.

This is true, AlphaFold predictions of the entire SpoIVFB-SpoIVFA-BofA complex seem inconsistent with the oligomeric interfaces observed in our cryo structures. For this and other reasons, we purposefully approached interpretation of the SpoIVFB oligomers observed here in a cautious fashion, and we are glad that the reviewer found these interpretations to be “balanced and diplomatic”. We also greatly look forward to structural analysis of the entire SpoIVFB-SpoIVFA-BofA complex.

12) The discussion might be strengthened by a few sentences about the RseP subfamily of site-2 proteases. These are significantly more broadly conserved among bacteria and play a key role in stress-response signal transduction pathways in gram-negative and -positive pathogens. The authors might emphasize that the regulated step in intramembrane proteolysis of anti-sigma factor substrates is ectodomain shedding. But specificity is likely imparted by beta-sheet augmentation without steric clashes. As such, RseP/RasP proteases are constitutive proteases but have specific substrate requirements.

Agreed, comparing the RseP-batimistat structure to that of other intramembrane proteases reveals a similar mode of ligand binding near the enzyme active site that involves interactions that mimic beta-sheet augmentation. We have included RseP in Supplemental Figure S10 and associated text in the first paragraph of the discussion, and added a few sentences to the end of the discussion, comparing RseP and SpoIVFB, mentioning their orthologs in pathogens, and noting their potential as therapeutic targets.

Reviewer #2 (Remarks to the Author):

In this study, Orlando et al report the cryoEM structure of the S2P SpoIVFB from B. subtilis in complex with its substrate pro- σ K as well as follow up biochemical experiments and simulations to support the experimental model and implicated mechanism of substrate recruitment and catalysis. This is a major leap for our understanding of S2Ps and intramembrane proteases more generally. Namely, all three intramembrane protease classes seem to use hybrid β -sheet formation between the substrate and enzyme as a driving force for luring the membrane-bound substrate into their hydrophilic active sites. The manuscript is generally well explained and thorough.

We thank the reviewer for their gracious comments. We also agree that the structures presented in the manuscript are “a major leap for our understanding of S2Ps and intramembrane proteases more generally”. Our cryo-EM structures presented herein represent a crucial piece of the puzzle linking beta-sheet augmentation mechanisms of substrate engagement to all three classes of intramembrane proteases.

The crosslinking (photo- and cys-) experiments have flaws that compromise interpretation, however. Some of the figures should be amended to convey more readily what is described in the narrative of the text. Finally, queries/curiosities for possible inclusion in a revised manuscript are listed below.

Structure

1. The statement that mjS2P and the SpoIVFB are similar yet have an rmsd of 3.5 Å is a contradiction. The overall topology is similar but extensive differences, e.g. in the pitch of TM3, TM6, plus a helix on the side is new, are apparent. Is the structural difference an artifact of mjS2P having been truncated or could there be a conformational change between apo and pro- σ K bound? (see also question about the simulations below). Perhaps more could be added to the manuscript about this.

It is true that while the overall topology of SpoIVFB is similar to that of mJ2P, differences between the two structures are also apparent. We note in the revision that “The topology of TM helices in SpoIVFB and mJ2P is the same, but the pitch of some TM helices is different (RMSD ~3.5Å overall).” Several factors confound a more detailed analysis of these conformational differences. First, the mJ2P structure was derived from a construct that was truncated after TM6 to remove the tandem CBS module of this protein. Second, the mJ2P structure contains zinc but lacks a bound proteolytic substrate. Lastly, mJ2P crystallized with two monomers in the asymmetric unit in an anti-parallel fashion (likely not biologically representative), but also in alternate open and closed conformations. Our comparison in the manuscript is between SpoIVFB and the “open” conformation of mJ2P. The degree to which variations in expression construct, zinc and substrate occupancy, crystal lattice contacts, oligomeric interfaces, and detergent interactions individually contribute to the structural disparities between the open conformations of SpoIVFB and mJ2P remains unclear. Despite the slight deviations between structures that are likely influenced by the above factors, the overall transmembrane region of both proteases has a very similar fold and the active site architecture is very well conserved.

However, it should be noted that a previous attempt at modeling an “open” and “closed” conformation of SpoIVFB based on mJ2P structures and evolutionary coupling analysis provided potential clues as to what conformational changes of SpoIVFB may occur (Ramírez-Guadiana, F.H. *et al. PLoS Genet* 14, (2018)). Based on the structures provided in the current manuscript, the models presented for the “open” and “closed” conformation of SpoIVFB by Ramírez-Guadiana, F.H. *et al.* are likely to be accurate. While such a closed-to-open transition seems plausible during substrate capture/activation, to our knowledge, biochemical evidence demonstrating conformational changes of SpoIVFB during activation or substrate processing is currently limited to inference from a single point mutation (F66A) and its effect on pro- σ^k processing *in-vivo* (Ramírez-Guadiana, F.H. *et al. PLoS Genet* 14, (2018)). This reference and interpretations for potential conformational changes in SpoIVFB are discussed in detail in the final paragraph of the revised manuscript.

2. Please include a zoom-in of the active site w/stick views of key residues so that protease aficionados can more easily deduce the contributors to the implied hydrolytic chemistry. Right now the most detail of the active site is Fig 4B. (Seems like the LMNG density can be omitted and the active site region be zoomed in).

A zoomed in view of the enzyme active site along with the proposed catalytic mechanism and key residues is included in Supplemental Figure S7A&B.

3. Figure 5A is challenging to interpret. Perhaps as surface representation of the different sections and/or electrostatic maps, combined with judicious use of stick for residues would help with this.

We agree that Figure 5A was slightly busy and difficult to interpret. In order to simplify this figure for the reader we have removed stick residues and labels for any residue that was not mutated in Figure 5B-D or referred to directly within the manuscript text. We hope that this simplifies the figure and points the reader to the relevant residues. Use of surface representation made the figure even more difficult to interpret.

4. Regarding methods, please include the rmsd of the final experimental models with the initial AF2 models used as initial models (and maybe rmsd of AF2 model and mJ2P as well, if pertinent)? As it becomes more common practice to use AF2 as a starting model for an experimental data set, I think this information is interesting to mention in manuscripts (was it mainly the side chains that needed adjusting or was more backbone rebuilding required to match density). Obviously, this is most interesting in cases where the starting and ending model differ notably, but either way this is a good data point since going

forward AF2 models for S2P orthologs will be updated as a result of this new structure and 'historical' AF2 structures will not be easily compared.

Indeed, the comparison of the AlphaFold2-Multimer prediction and final cryo-EM model is interesting. AlphaFold predicts the overall fold of SpoIVFB and Pro- σ^K , but some differences are apparent (RMSD between SpoIVFB:SpoIVFB = 2.6Å, RMSD between Pro- σ^K :Pro- σ^K = 7.3Å). As can be seen in the image above, the largest differences between the AlphaFold model and the cryo-EM model are in the configuration of the pro-sequence of Pro- σ^K and the membrane reentrant loop that engages the substrate through β -sheet augmentation. In the AlphaFold model the pro-sequence is in an alpha-helical conformation, whereas in the cryo-EM structure it is captured in the active site in a beta-sheet configuration. In this instance, AlphaFold did a remarkable job of predicting the interactions between Pro- σ^K and the interdomain linker of SpoIVFB, but did not capture the mechanism of β -sheet augmentation, which was only revealed through the experimental cryo-EM map. This particular example highlights both the power of AlphaFold and also the biological insights that can be further gained through experimental structure analysis.

We have included the following statement in the methods...

“A comparison of the final cryo-EM derived atomic model and the initial AlphaFold model reveals an overall RMSD of 2.6Å between SpoIVFB:SpoIVFB and an RMSD of 7.3Å between Pro- σ^K :Pro- σ^K . The largest difference between the experimental cryo-EM structure and the AlphaFold prediction of the SpoIVFB:Pro- σ^K complex lies in the configuration and positioning of the Pro- σ^K pro-sequence (alpha-helical in AlphaFold prediction, and beta-sheet configuration in the cryo-EM derived model).”

Biochemistry/Simulations

1. It's remarkable that many point mutations have no effect on activity, but in the case where enzyme activity was abrogated e.g. Y204A/R208A, it would seem prudent to include blots showing that the mutants were actually expressed and that loss of activity observed is not due to enzyme misfolding and degradation.

In the cleavage assays presented in Figure 4D and 5B we included a western blot showing expression of both Pro- σ^K / σ^K and SpoIVFB. These blots demonstrate that overall expression levels of the variant proteins were comparable to wild-type protein, indicating that any differences in activity are not due to gross enzyme misfolding or degradation. Figure 5B clearly shows that the Y204A, R208A, and Y204A/R208A variants of SpoIVFB are expressed at a similar level to wild-type protein. It is important to note that the SpoIVFB and Pro-

σ^K/σ^K blots shown in the main figure were cropped from the same western blot to save space in the main figures. All uncropped complete western blots are provided in the “Source Data” file.

2. None of the photo- or Cys-crosslinking blots have a ladder or another way to show that the crosslinked species are the two proteins of interest and not multimers of one component. This is a major flaw of the paper.

All of the disulfide crosslinking blots indicate the migration of protein molecular weight (MW) markers (Fig. 5F and Suppl. Fig. S8B-E). We have clarified this in the figure legends. The species migrating at ~45 kDa is the complex between SpoIVFB (~34 kDa) and Pro- σ^K (~16 kDa), as explained further below. Only SpoIVFB (not Pro- σ^K) is FLAG-tagged and detected in the anti-FLAG blots. SpoIVFB dimer migrates at >60 kDa, as observed in some experiments with 1 mM Cu²⁺(phenanthroline)₃ oxidant for 60 min to promote disulfide crosslinking (e.g., Olenic *et al. J Bacteriol* (2022)), but we observed very little to no SpoIVFB dimer formation in the experiments reported herein with 1-100 μ M Cu²⁺(phenanthroline)₃ for 5 min (see uncropped western blots provided in the “Source Data” file).

In the anti-pentaHis blot in Suppl. Fig. S8C, both SpoIVFB and Pro- σ^K are His-tagged and detected in the two lanes labeled “+His”. The two lanes labeled “ Δ His” in the same blot demonstrate the presence of Pro- σ^K in the ~45 kDa species, since SpoIVFB in those samples lacks a His-tag. Formation of the ~45 kDa species reflects disulfide crosslinking since it is observed in the sample treated with oxidant (Cu) but not reductant (DTT). Since Pro- σ^K has a single Cys (N53C) in this experiment, disulfide crosslinking can form a dimer of Pro- σ^K , and a very faint species migrating at ~30 kDa (as expected for Pro- σ^K dimer) is seen in the longer exposure of the oxidant-treated sample, but disulfide crosslinking cannot form a trimer of Pro- σ^K , so we conclude that the ~45 kDa species is the complex between SpoIVFB and Pro- σ^K .

For the photocrosslinking blot we have also added labels for the approximate locations of molecular weight markers in Figure 5D and the Source Data file.

3. Full blots for both photo- and Cys-crosslinking blots should also be presented in the supplemental data, presumably this is required by the journal.

Indeed, this is required by the journal. Full uncropped blots for all data presented in the manuscript have been included in the Excel “source data” file.

4. Please move Supp Fig. S8B to the main Figure 5 and amend to highlight where the strongest crosslinks are found in the structure, e.g. Y215/N53. Otherwise, the text is hard to follow.

We have moved Supplemental Figure S8B to main Figure 5E, and highlighted only the strongest crosslinks that are observed in Figure 5F. We agree, this greatly improves readability of the text describing the disulfide crosslinking.

5. I will leave the quality of the simulations to an expert reviewer in that methodology, but interpreting the simulations as presented is not straightforward with the images provided in Fig. 6A. Perhaps an electrostatic surface representation would complement the apparent trajectory of the waters into the active site. Further zoom into the active site with the substrate bound alongside the waters (and labeling all residues even after 90 ° rotation please) might assist with interpretation.

As suggested by reviewer #3 below, we have significantly expanded our MD analysis with additional simulations, and have provided additional analysis from four independent replica simulations. Several figures have been updated to reflect these additional experiments and analysis. Figure 6 has been updated to show average occupancy maps (Fig. 6A & B) for water, providing a sense of where water penetrates throughout the simulation. Figure 6C & D now show a zoom view of the active site region with substrate and water visible after $\sim 120^\circ$ rotation and all active site residues and TM helices labeled. We have also included Suppl. Fig. S9G-L which provide analysis of water permeation into the active site region, and a Supplemental Video 1 showing water permeation into the active site. All simulation data have also been uploaded to the public repository Zenodo so that readers can visualize the trajectories in 3-dimensions.

6. Do the authors think water binds first and the substrate displaces waters already present? Is this an additional driving force (entropy/water release) that contributes to substrate binding in the active site?

This is an excellent question that we simply cannot answer with confidence at this time. As mentioned in the response to point #1 above, it is currently unclear whether SpoIVFB undergoes conformational transitions during the substrate engagement process. Furthermore, it is unclear what conformation SpoIVFB adopts when it is in the inhibited complex with SpoIVFA and BofA. While it is tempting to speculate that increased water entropy could be a driving force contributing to substrate binding, without knowing the conformation of SpoIVFB before the substrate engagement process, it is difficult to speculate confidently on the contributions of water. If SpoIVFB indeed adopts a closed conformation similar to that observed for mjS2P before engaging substrate, then one could expect that release of water from the active site upon substrate engagement may not be a major driving force. On the other hand, if SpoIVFB is held in an open conformation prior to substrate engagement and water fills the empty active site, then release of water from the active site upon substrate binding seems more plausible as a driving force. Without knowledge of the conformation of SpoIVFB before engaging Pro- σ^K , it is difficult to speculate with confidence on the entropic contributions of water to substrate engagement.

Other Comments

1. I am not concerned about the structure being of an apo SpoIVFB with pro- σ^K . Surely the authors have tried spiking with Zn at the time of induction. If the hypothesis is right, namely, Zn is labile and lost during purification, presumably you would get more cleaved pro- σ^K and less intact complex if you spiked with Zn. A comment along these lines could mitigate the caveat about the study early in the paper.

Zinc does not appear to be limiting for SpoIVFB activity during induction. Addition of zinc to cultures does not change the cleavage ratio *in vivo* (i.e., 80-90% of the co-expressed Pro- σ^K is cleaved to σ^K), as shown in the newly added Suppl. Fig. S1A and mentioned in the revised text near the beginning of Results. We do not understand what limits SpoIVFB activity *in vivo*. If zinc is labile and lost from SpoIVFB *in vivo*, then one might presume that zinc addition to cultures would increase SpoIVFB activity and the cleavage ratio. However, our hypothesis that zinc is lost during purification stems from our *in vitro* results. Purified WT SpoIVFB:Pro- σ^K complexes lack zinc and as stated in the Results, our “Attempts to reconstitute *in vitro* proteolytic cleavage of Pro- σ^K by supplementing purified WT SpoIVFB:Pro- σ^K complexes with Zn²⁺ (by adding Zn-acetate or Zn-chloride) have thus far been unsuccessful.” In the Discussion, we hypothesize that “It is possible that the catalytic zinc ion is lost during protein purification, and once Pro- σ^K binds to a zinc-depleted SpoIVFB the pro-sequence blocks the zinc-coordinating residues and prevents future complexation with exogenously supplied zinc.”

2. Based on the structure, do the authors have any insight into S2Ps that contain other domains, e.g. PDZ? It might be nice to broaden the scope of the findings for SpoIVFB to others S2Ps in the discussion.

See response to reviewer #1 point #12 above. We have expanded the discussion and added a supplemental figure (Suppl. Fig. S10) to include proteases with additional domains, such as RseP.

3. One of the interesting findings specific to S2Ps is that the interdomain linker plays a major role in mediating SpoIVFB -pro- σ^K interactions. How conserved is this region, say, across bacterial S2Ps with a similar overall domain architecture to SpoIVFB? If conservation is low, then there might be different ways different types of S2Ps engage their substrates, which would be worth mentioning as a limitation of the present study but worthwhile for future follow up.

A figure showing sequence conservation among SpoIVFB orthologs plotted onto the atomic model is included in Supplemental Figure S7C. This figure was derived from generating a multiple sequence alignment using the “clustered nr” BLASTP algorithm which has more taxonomic depth than standard BLASTP searches. Approximately 100 sequences from orthologous species were used to generate Supplemental Figure S7C, which indicates that the C-terminal region of the interdomain linker is quite well conserved among proteins with a similar domain architecture to SpoIVFB. Interestingly, this is also one of the regions where we observed the strongest disulfide crosslinks (Figure 5 E&F). To highlight these observations, we labeled the interdomain linker in Supplemental Figure S7C and added a sentence to the Discussion – “Sequence conservation among SpoIVFB orthologs indicates that the C-terminal region of the interdomain linker is quite well conserved (Suppl. Fig. S7C) and we observed strong disulfide crosslinks between this region and Pro- σ^K (Fig. 5E&F), suggesting that interaction between the C-terminal region of the interdomain linker and substrate is a conserved feature of substrate engagement for SpoIVFB and its orthologs.”

4. As alluded to in several places above, the finding that three classes of IPs--SpoIVFB, RseP+RhoB, and γ -secretase—engage substrates by forming a hybrid β -strand between substrate and enzyme is one of the most powerful broader implications of this paper. One of the fundamental questions surrounding IPs has been how a transmembrane/mostly hydrophobic helix would find its way into a chemically-incompatible hydrophilic active site for cleavage, and now it is apparent! This commonality is mentioned in the first paragraph of the discussion but I think the paper would benefit from further expansion. To what extent might the driving force for helix unwinding be to form a hybrid β -strand/enthalpic vs displacing waters/entropic? If all IPs use this ‘mechanism’ then why are some IPs processive and others specific?

Based on the comment from reviewer #1 above we have included Supplemental Figure S10, highlighting the commonality of the β -sheet augmentation mechanism between intramembrane proteases SpoIVFB, RseP, GlpG, and γ -secretase. We have also expanded the discussion to further emphasize the conserved beta-sheet augmentation mechanism utilized across intramembrane proteases of diverse classes.

While beta-sheet augmentation seems to be a common substrate engagement mechanism operating across broad classes of intramembrane proteases, based on the data currently available for SpoIVFB we feel it is premature to expand upon the enthalpic and entropic contributions for helix unwinding and hybrid β -strand formation. As noted above in response to point #6, we currently do not know the conformation of SpoIVFB prior to substrate engagement, or whether or not water fills the active site prior to engagement of substrate. Without this critical information it is difficult to confidently speculate on the relative thermodynamic contributions to substrate binding, and further investigation into this area is warranted in future studies of SpoIVFB and related intramembrane proteases.

With respect to processive vs. specific intramembrane proteases, presenilins and signal peptide peptidases are aspartyl intramembrane proteases that exhibit carboxypeptidase-like processive/successive cleavage of

substrates after an initial endopeptidase-like cleavage. The most intensively studied of these enzymes, γ -secretase, has a presenilin catalytic subunit that first executes endopeptidase-like cleavage of the amyloid precursor protein (APP), followed by processive C-terminal cleavages. Helix breaking of the substrate for the initial cleavage involves the beta-sheet augmentation mechanism (Zhou *et al. Science* (2019)). Recent work suggests that conformational flexibility of presenilin is important for both the initial endopeptidase-like cleavage and the first carboxypeptidase-like cleavage (Devkota *et al. Cell Reports* (2024)). Thus, enzyme flexibility appears to be important for processive cleavage. Perhaps specific intramembrane proteases are less flexible, but we think this is too speculative for inclusion in this manuscript.

During the preparation of this revised manuscript three new PDBs were released for the γ -secretase complex bound to APP-C99 (PDB = 8X54), A β -49 (PDB = 8X52), or A β -46 (PDB = 8X53), which depict positioning of substrates during processive (successive) peptide cleavage reactions. We have included Suppl. Fig. S10I-K to demonstrate the substrate positioning in these new structures. As can be seen from these new structures, in the processive γ -secretase complex the substrate reengages the enzyme through successive reformation of an additional beta-strand following each cleavage reaction. A difference between enzymes that appear to have processive cleavage activity (ie: γ -secretase) versus those that catalyze cleavage at a single defined site (ie: SpoIVFB and RseP) lies in the positioning of the enzyme active site relative to the induced beta-strand. In non-processive proteases (SpoIVFB and RseP) the cleavage site lies directly in the middle of the beta-strand of the substrate, whereas in a processive protease (γ -secretase) the active site is positioned near a loop that lies just beyond the extended beta-strand configuration of the substrate (Suppl. Fig. S10I-K). Whether or not these are defining features of all processive versus specific proteases awaits further structural and biochemical work.

Reviewer #3 (Remarks to the Author):

M. A. Orlando and colleagues have determined the high-resolution structure of SpoIVFB in complex with its proteolytic substrate Pro- σ K using cryo-electron microscopy single particle analysis. SpoIVFB is a member of the S2P family of intramembrane metalloproteases, characterized by a C-terminal CBS domain. However, in the existing resolved structures of other S2P family intramembrane metalloproteases, the CBS domain has consistently been omitted for simplicity and has never been resolved within a complex.

This study presents a detailed molecular view of substrate engagement and positioning within an S2P intramembrane metalloprotease, offering insights into the mechanism of Pro- σ K cleavage facilitated by SpoIVFB. The authors further enriched these findings with a comprehensive mutational analysis and protein cross-linking experiments to elucidate the protein-protein interactions essential for the precise positioning of Pro- σ K during intramembrane proteolysis. Additionally, they employed molecular dynamics simulations to assess the stability of the structure within a model membrane, revealing the formation of a water-filled cavity.

The manuscript is well-written, featuring clear, beautifully presented, and straightforward to comprehend figures. I thoroughly enjoyed engaging with this high-quality work, and it merits consideration for publication in Nature Communications, given its novelty and the fascinating insights it provides.

Nonetheless, the section on molecular dynamics simulations should be improved. The authors have conducted a single 250 ns molecular dynamics simulation to visually demonstrate the influx of water into a cavity and the potential binding site for a PE lipid. This is a limited approach to exploring these complex

interactions.

Regarding the simulations, I appreciated the author's effort to include them. I believe the depiction of a water channel adds a compelling element to the narrative, suggesting a plausible mechanism by which water facilitates the hydrolytic cleavage of the Pro- σ^K peptide backbone. However, the presentation is limited to snapshots from the MD simulations rather than a comprehensive data representation. Moreover, the study relies on a single MD simulation with an unsuitable membrane model composition.

1. I would advise the author to conduct at least three independent simulations (in this case building three times from CHARMM-GUI) and to calculate average properties such as hydration numbers (the number of water molecules over time within the cavity) and contact occupancy between water molecules and residues. Additionally, analyzing the interaction energy would provide a detailed characterization of average properties and the hydrophilic route. Furthermore, I have concerns regarding the choice of model membrane utilized in this study. The authors opted for a 3:1 mixture of POPE:POPG. It is known that PE is not a bilayer-forming lipid, and at concentrations greater than 60%, vesicles are not stable in experiments due to induced defects and phase transitions. Additionally, I question the rationale behind using PG (e.g., I would have chosen POPC instead) or of this specific composition in general. Is there a specific reason for its inclusion? How did the authors arrive at this particular membrane composition?

We thank the reviewer for pointing out the limitations of the single MD trajectory and concerns over the bilayer composition presented in our initial manuscript. With regard to membrane composition, the POPE:POPG mixture was chosen as a rough mimetic of bacterial membrane composition. Common extracts of *E. coli* lipids that are commercially available (Avanti Polar lipids) contain roughly 67% PE and 23% PG lipids (*E. coli* polar extract) or 56% PE and 15% PG lipids (*E. coli* total extract). In many studies throughout the literature an approximate 2:1 mixture of PE/PG is utilized to mimic a bacterial membrane in liposome-based studies. For a particularly nice example utilizing such liposomes please see...

Chadda R, Lee T, Mahoney-Kruszka R, Kelley EG, Bernhardt N, Sandal P, Robertson JL. A thermodynamic analysis of CLC transporter dimerization in lipid bilayers. *Proc Natl Acad Sci U S A*. 2023 Oct 10;120(41): PMID: 37788312

However, we acknowledge that POPE:POPG mixtures can also demonstrate unusual aggregation, phase transition temperatures, etc... depending on experimental conditions, and may not be the ideal composition for molecular simulation. To our knowledge, one of the most comprehensive analyses of *Bacillus subtilis* membrane lipid composition and simulation properties can be found in...

Nickels JD, Chatterjee S, Mostofian B, Stanley CB, Ohl M, Zolnierczuk P, Schulz R, Myles DAA, Standaert RF, Elkins JG, Cheng X, Katsaras J. *Bacillus subtilis* Lipid Extract, A Branched-Chain Fatty Acid Model Membrane. *J Phys Chem Lett*. 2017 Sep 7. PMID: 28825491

In the manuscript above the authors reveal through GC-MS based approaches that the *Bacillus subtilis* membrane is composed of roughly 49% PE, 28% DAG, 8% PG, and 3% cardiolipin lipids with varying acyl chain lengths and positions of iso-methyl groups. Based on this previous characterization of *B. subtilis* lipids we constructed a new model membrane for SpoIVFB:Pro- σ^K simulations composed of 53% POPE, 30% DAG (16:0, 18:1), 14% POPG, and 3% cardiolipin (16:0, a15:0, 16:0, a15:0) to resemble the distribution of lipids previously found in *B. subtilis* membranes. Importantly, we independently assembled four simulation boxes with SpoIVFB:Pro- σ^K embedded in this new membrane composition with independent runs through CHARMM-GUI. Each system was independently equilibrated and simulated without constraints for 250ns, totaling 1

microsecond of total simulation time. RMSD values from each independent simulation reveal similar overall deviation from the starting structure for each independent run (Suppl. Fig. S9C-F), and we calculated the average penetrance of water within the vicinity (7Å) of SpoIVFB catalytic residue E44 over time (Suppl. Fig. S9G-L). The methods, results, and analysis of these four independent simulations have been added to the appropriate places in the manuscript and figures. This additional analysis reveals a similar scale and timeframe of water penetration into the SpoIVFB active site in each replicate simulation.

We thank the reviewer for their constructive criticism on the lipid composition and MD analysis. We believe that the more representative membrane composition and additional replicate simulations significantly strengthen our manuscript. We updated Figure 6 and added Supplemental Figure S9 to display the new results.

2. For these reasons, I recommend that the authors explore alternative membrane compositions and if possible, more than one. The protein may interact differently with different membrane environment, potentially impacting protein orientation and the water channel formation. For each membrane composition tested, a minimum of three molecular dynamics simulations should be conducted to ensure robust and statistically reliable results.

We appreciate the reviewers comment, and acknowledge that further simulations in different lipid environments could reveal interesting insights. However, at this time we have completed 1 microsecond of total simulation time in what we believe to be a lipid bilayer that closely resembles the lipid composition of *B. subtilis*. Furthermore, the additional four replicate simulations in a POPE:POPG:DAG:CL bilayer show essentially the same results as the original simulation in a POPE:POPG bilayer. For these reasons we feel that additional simulations in bilayers with alternate lipid compositions are outside the scope of the current manuscript, and will require significantly expanded *in-silico* studies. Our intent with the simulations was simply to demonstrate a plausible pathway for water to enter the vicinity of the enzyme active site, and to probe the possibility of lipids positioning above the membrane reentrant loop of SpoIVFB (discussed more below), which we believe has been achieved with the additional replicate simulations now supplied.

3. The authors' assertion that the protein traps a POPE lipid molecule lacks statistical evidence since it is based solely on a single MD simulation where the lipid immediately migrates to a specific site. To substantiate this claim, the authors should conduct multiple independent simulations where the lipid is initially positioned away from the pocket. For instance, after the lipid moves to the pocket, the author could remove it and see if another lipid returns in, offering a more rigorous test of its affinity. Additionally, I expected the authors to delve into characterizing the nature of the lipid-protein interaction—whether it is dependent on the lipid's tail or head or merely driven by hydrophobic interactions. This could be achieved through detailed analyses such as contact occupancy and interaction energy, providing deeper insight into the interaction dynamics.

It is important to note that our intent was not to imply that SpoIVFB traps a specific lipid (POPE) over the membrane reentrant loop near the enzyme active site, but rather to demonstrate that the general environment is conducive to accommodating the hydrophobic lipid acyl chains similar to that observed for LMNG in the cryo-EM structure. Indeed, a more robust analysis from replicate simulations would provide stronger support for such a claim. In each of the four new replicate simulations, independent runs of CHARMM-GUI generated four independent simulation systems with different membrane lipids initially placed near the membrane reentrant loop. In each of the simulations a lipid molecule rapidly traverses to the region over the SpoIVFB membrane reentrant loop and remains in this vicinity for the bulk of the simulation time. In simulation 1, 3, and 4 the lipid happens to be a DAG, which lacks a phospho-headgroup. In simulation 2 the lipid is a POPG. The fact that POPE (simulation in original manuscript), POPG, and DAG all seem to position in

similar places as LMNG observed in our cryo-EM structure suggests that the headgroup of the lipid has little to no effect on stabilizing the lipid interaction, which is rather driven largely by hydrophobic interaction between the lipid acyl chains and the hydrophobic surfaces of SpoIVFB. This situation is also demonstrated by the figure below of the cryo-EM structure with the SpoIVFB surface colored according to hydrophobicity as implemented in UCSF Chimera. As can be seen in this representation, the surface of SpoIVFB in the region occupied by the acyl chains of LMNG (magenta sticks) is largely hydrophobic, and transitions to hydrophilic near the maltose headgroups. We have added the left part of the figure below as Figure 6E and we have expanded the relevant section of results to include the new simulations.

As the reviewer suggests, it would be interesting to “remove” the lipid and see if another lipid returns. Rather than “removing” the lipid we took the approach of setting up an additional simulation of SpoIVFB:Pro- σ^K embedded in the new membrane composition, and we utilized the “generate pore water” feature of CHARMM-GUI to initially place waters above the membrane reentrant loop (see first panel figure below) and in the general pocket that is occupied by LMNG in the cryo-EM structure or lipid in the first 4 simulations. This 5th system then underwent 250 ns of unrestrained simulation. As can be seen in the trajectory snapshots below, the “pore water” that was initially placed above the membrane reentrant loop during system setup rapidly (within ~60 ns) is displaced as lipid acyl chains move into the hydrophobic pocket behind the membrane reentrant loop and between TM1 and TM2 (i.e., the region occupied by LMNG in the cryo structure). In the figure below lipids are only shown in the final snapshot (250 ns) for clarity, although they begin to penetrate the region occupied by “pore water” within ~60 ns. Unlike in the other simulations, soon after releasing equilibration restraints the individual β -strands of the membrane reentrant loop separate from one another, allowing a larger influx of water into the general vicinity of SpoIVFB residue E44, resulting in significantly more waters (~10 versus ~5) residing near this catalytic residue near the end of the simulation (see the plot below). Due to the unusual (likely unrealistic) separation of the membrane reentrant loops in this final simulation, we did not include data from this simulation in calculation of water occupancy near the SpoIVFB active site (Suppl. Fig S9G-K), or in discussion/analysis of the different trajectories. Nevertheless, this simulation shows similar behavior as the other four replicates in that the hydrophobic pocket occupied by LMNG in the cryo-EM structure (see figure above) becomes filled with lipid acyl chains even if the pocket is originally solvated by water. Although this final simulation is not included in our analysis/discussion, we have uploaded the relevant trajectory files to Zenodo along with the other simulation results such that they are freely available to any readers of the manuscript.

We acknowledge that deeper analysis of interaction energies between different lipids and SpoIVFB would likely provide interesting insights into potential lipid preferences. However, simply extracting electrostatic and van der Waals force field energy components for the lipid-protein interactions is unlikely to provide meaningful estimates of a binding free energy, which is one of the critical parameters for evaluating potential lipid preferences. Such analyses require significantly expanded and technical simulations that are outside the scope of the current manuscript. Rather, our primary goal was to demonstrate plausible routes for water access to an active site buried within a membrane and to show that lipids (and detergent) appear to have a preference for binding behind the membrane reentrant loop where LMNG is present in the cryo-EM structure. The cryo-EM structure indicates that the surface of SpoIVFB is largely hydrophobic in this region (Fig. 6F), allowing LMNG to bind relatively stably. Our simulations provide further confidence that this region is also likely occupied by hydrophobic acyl chains in a membrane context. While further work is certainly warranted to explore potential roles of lipids and water permeation pathways in more detail, we believe that the current

simulations and expanded analysis largely support the conclusions and discussion presented in the revised manuscript.

4. Furthermore, the methods section lacks references for the CHARMM36 force field, thermostat, barostat, and algorithms for calculating electrostatic interactions. Additionally, the water model used in the simulations is not specified. Furthermore, data about the molecular dynamics simulations are not accessible for reproducing or visualizing the trajectory. I recommend making at least the minimum required files for reproducing the simulations publicly available. Numerous platforms, such as Zenodo.org, can provide a Digital Object Identifier (DOI) for this purpose.

Thank you for pointing out these deficiencies. We have added citations for the CHARMM36m force field, thermostat (integrator), barostat, and PME for long range electrostatic interactions. We have also now indicated in the methods that the systems were solvated using the TIP3P water model. We have also uploaded all relevant files needed to reproduce and visualize the trajectories to Zenodo such that the information is freely available for inspection.

Reviewer #4 (Remarks to the Author):

The manuscript entitled "Substrate Engagement by the Intramembrane 1 Metalloprotease SpoIVFB" by Orlando et al. mainly focuses on cryo-EM-based structural characterization of Metalloprotease SpoIVFB bound to the substrate Pro- σ^K from Bacillus subtilis, which is one of first cryo-EM structure of SpoIVFB bound Pro- σ^K . However, this structure was characterized with full-length Pro- σ^K . From these structural studies, it is not clear how enzymatically active SpoIVFB was able to bind with full-length Pro- σ^K , which is a substrate of SpoIVFB.

It should be noted that the structures presented in the manuscript are not simply "one of the first cryo-EM structures of SpoIVFB bound to Pro- σ^K ", they are the only experimentally derived (i.e., not *in-silico* prediction) structures currently available for both SpoIVFB and Pro- σ^K , and furthermore are the only experimentally-derived structures of any site-2 protease engaged with a proteolytic substrate. As described in the manuscript, the reason WT SpoIVFB bound Pro- σ^K without cleaving it is the absence of the catalytic zinc ion in the purified complex.

The authors also performed cryo-EM structure calculation of mutated SpoIVFB. Additionally, mutational and protein crosslinking analyses pointed to several key amino acid residues of protein-protein interactions between SpoIVFB and Pro- σ^K . The authors performed a good amount of work in this manuscript, which covers protein expression & purification, mutation, crosslinking studies, and cryo-EM to calculate the 3D structure. Authors resolved high-resolution structures of Wt and mutated SpoIVFB with Pro- σ^K . This manuscript was well-written, and overall, the representation was good. However, more clarifications are required during the data represented.

1. *The major drawback of this work was that Pro- σ^K is a substrate of SpoIVFB, which should be proteolytically cleaved by SpoIVFB. However, the authors resolved the full-length structures of SpoIVFB with Pro- σ^K . This raises an important question: Did the authors select inactive SpoIVFB for structure calculation? If authors were aware of this, why did they not take suitable steps to purify the active protein? How is this structural information from inactive proteins relevant?*

As elaborated upon in the manuscript, the SpoIVFB complex that is isolated from *E. coli* is devoid of the catalytic zinc ion, which renders the enzyme inactive. Despite extensive efforts we have been unable to reconstitute *in-vitro* proteolysis using the purified SpoIVFB:Pro- σ^K complexes utilized here for cryo-EM. An

entire paragraph of the discussion (starting on line 379 in the revised manuscript) was dedicated to elaborating upon the implications of the lack of the catalytic zinc ion and efforts to reconstitute enzymatic activity. Although the purified protein complex lacks the catalytic zinc ion, the experimental structures still provide valuable information on the interaction interfaces between SpoIVFB and Pro- σ^K , positioning of the substrate over the enzyme active site through β -sheet augmentation, and the orientation and role of the interdomain linker and CBS domain in substrate capture. Although this information was gleaned from structures that lack the catalytic zinc ion, the site-directed mutagenesis, crosslinking, and simulation data all suggest that the structure observed with cryo-EM is consistent with a configuration of enzymatically active protein *in-vivo*.

2. *The authors performed photo-crosslinking and disulfide crosslinking. Samples were used to perform immunoblot analysis; It may provide positive results due to disulfide crosslinking. The author should clarify how crosslinking affects their immunoblot and other biochemical analyses. I hope the authors did not use this crosslinked protein for their cryo-EM analysis. One possibility is that the authors resolved the full-length Pro- σ^K with SpoIVFB due to crosslinking.*

Disulfide and photo-crosslinking assays were utilized to verify that the interaction interfaces between SpoIVFB and Pro- σ^K observed in the cryo-EM structures are relevant to the interactions that mediate substrate capture and cleavage *in vivo*. Both of the crosslinking assays were performed in live *E. coli* co-expressing SpoIVFB and Pro- σ^K (a system where robust catalytic activity is observed), and the results provided validation that the cryo-EM structures are representative of the enzyme-substrate complex encountered *in vivo*. Importantly, none of the protein preparations used for cryo-EM analysis underwent crosslinking. The SpoIVFB:Pro- σ^K complexes used for cryo-EM were isolated through a common two-step IMAC and gel-filtration purification scheme before being immediately frozen on cryo-EM grids.

3. *I am seriously concerned about cryo-EM data processing. Author mentioned in Supplemental Figure S2A&B that authors observed several oligomeric states of Pro- σ^K with SpoIVFB; oligomeric states, including dimers, trimers, and tetramers of SpoIVFB:Pro- σ^K (Suppl. Fig. S2A&B) were observed in 2D class averages. How did they identify dimers from tetramers and trimers? Generally, from the top view, it is easy to identify tetramer and trimer, but I am unable to see any dimeric top views. However, isolating dimers from trimers and tetramers from the side views of 2D averages is challenging. How did the authors isolate side views of this complex? It will be better to run 2D averages of the individual 3D model data sets and show that particles were segregated based on dimer, trimer, and tetramer.*

Indeed, data processing for a sample with several oligomeric states present in initial 2D classification is challenging. During initial rounds of 2D classification it is easy to identify trimers from dimers/tetramers with the top view averages. It is also important to note that in the top view 2D averages of trimers, one of the three Pro- σ^K is consistently weaker than the other two, suggesting that compositional heterogeneity is still present within these particle populations. However, distinguishing a side view of a dimer from a similar tetramer view can be difficult or impossible to the human eye. For this reason, we purposefully did not apply any symmetry during cryo-EM data processing until the final refinement of the final classified particle subset. As can be seen in Suppl. Fig S2C the initial round of *ab-initio* reconstruction and heterogeneous refinement with C1 symmetry clearly separated a class representative of a dimer (grey volume) and a tetramer (cyan volume), along with two other “junk” classes that likely contain some of the top view trimer particles that otherwise do not reconstruct into a homogenous volume. Only particles corresponding to the tetrameric class were selected for further rounds of 3D classification (without applied symmetry). 2D averages derived from only the particle sets that went into the final 3D refinement are shown in main Figure 2B&D.

4. *How did the authors generate the initial model? In Figure S2C, they pointed out that "attempts to*

reconstruct the SpoIVFB:Pro- σ^K trimers were unsuccessful, likely due to severe orientation bias of this particular species". Could they provide the number of particles and angular distribution of the trimeric state?

Supplemental Figure S2 and S5 have been updated to indicate that *ab-initio* reconstruction in CryoSPARC (without any applied symmetry) was used to generate the initial models, and this is also indicated in the methods section. The population of "trimeric" complexes identified from top view 2D averages is only ~23,000 particles, and as noted above, deciphering side views of different oligomeric species is difficult. Due to the limited particle number, likely biased angular distribution (selection of only top views), and inability of *ab-initio* reconstruction or heterogeneous refinement to reveal a clear "trimer" class of particles, there is no reliable 3D reconstruction of a trimeric species to report an accurate angular distribution.

5. Figure 2d, 2nd, and last class average showed only one extra density from the membrane part. I can expect this 2d came from the final 3d structure, and a small extra density should be for the non-member part of SpoIV and Pro- σ^K , but suppl Fig 2b, the last class avg (blue circle) mentioned it is dimer... this is contradicting author's claim.

Main Figure 2D shows the final 2D averages calculated from the dimeric E44Q variant dataset. The reason there is only one "non-membrane part" (soluble region of Pro- σ^K) in the averages the reviewer points out is because the orientation of the dimer in these classes is such that the individual Pro- σ^K subunits are stacked upon one another along the z-axis (i.e., parallel to the axis of the electron beam). The 2nd and last averages show a side view of the dimer as would be seen if both Pro- σ^K were stacked upon one another in the axis parallel to the electron beam. If one were to take the first 2D average in main Figure 2D and rotate the particle ~90° along the x-axis, the second average in this figure would be the result. As indicated in Supplemental Figure S2B, the dimeric species is also present as a small population in the WT SpoIVFB:Pro- σ^K dataset. It is important to note that the 2D averages presented in Suppl. Fig. S2B are from an early stage of data processing (before any heterogeneous refinement/3D classification). These dimers were effectively filtered out of the dataset during subsequent *ab-initio*/heterogeneous refinement/3D classification.

6. The authors showed in Figure 3 and Supplemental Figure S7 and Figure 4 the structure of the SpoIVFB intramembrane protease and the zinc identified in the mjs2P structure. The binding of the pro-sequence of Pro- σ^K to SpoIVFB showed LMNG detergent. However, this 3D structure should be surrounded by LNMG. Why did the authors observe only one LNMG density?

In order to clearly resolve a detergent molecule in a cryo-EM map the detergent needs to be well ordered through specific binding to the protein. The transmembrane region of the SpoIVFB:Pro- σ^K complex is surrounded by disordered detergent as indicated by the low resolution map showing the detergent micelle in main Figure 2A&C (transparent grey outline). The detergent micelle surrounding the transmembrane helices is also clearly present in 2D averages of the complex. Detergent molecules surrounding the transmembrane region are largely disordered (i.e., not positioned the same in every individual particle) within the larger micelle mixture, whereas the LMNG bound over the active site of SpoIVFB is particularly well ordered and gives rise to defined density in the final cryo-EM map.

7. Authors should show the atomic model fitted map and point to the Zn density as well as LNMG density. It may have an important role if this LNMG density is located inside the TM domain. Did the authors perform any experiments, or did they have any explanations for the role of LNMG and lipids inside the TM? Is there any evidence that LNMG or lipid is required for activation?

As indicated in the manuscript the SpoIVFB:Pro- σ^K complex is devoid of zinc (explaining the lack of catalytic activity). We agree, that the lipid/LMNG located inside the TM domain may play an important role, and for this reason expanded upon our results from molecular simulation showing a similar positioning of membrane lipid as that observed for LMNG in the cryo-EM map. At this time, we are not aware of any reports demonstrating that a particular lipid species is involved in SpoIVFB regulation or catalysis, but the results presented in the current manuscript suggest that this may be an important topic to consider in future work.

8. The authors proposed, "In total, the physical interaction between monomers in the WT SpoIVFB:Pro- σ^K tetramer is quite limited ($\sim 488\text{\AA}^2$ for interface 1, $\sim 510\text{\AA}^2$ for interface 2), which makes for fragile interactions between SpoIVFB:Pro- σ^K monomers that are largely mediated by lipid/detergent....." However, the authors did not perform any proper experiments to support their claim.

The cryo-EM experiments themselves provide direct insight into the interaction surfaces between SpoIVFB:Pro- σ^K monomers, which were observed to be quite small ($\sim 500\text{\AA}^2$), and in several points mediated by protein-lipid/detergent interaction. In order to test the role of lipid/detergent in mediating oligomer formation the SpoIVFB:Pro- σ^K complex would need to be purified in a variety of different detergents and solubilized at different detergent:protein ratios, and then analyzed for distribution of oligomeric species. As the different oligomeric species (dimer/tetramer) run at similar elution volumes on an S200 gel-filtration column, it is difficult to perform experiments such as SEC-MALS where the contribution of individual oligomeric species would need to be deconvoluted from each other running in the same elution peak. Rather, our intent with the comment above was simply to note that in the oligomeric species much of the interaction between protomers is mediated by detergent/lipid. These fragile interaction interfaces can likely lead to prep-to-prep variability in the population of oligomeric states. We have amended the sentence to read as the following...

"In total, the physical interaction between monomers in the WT SpoIVFB:Pro- σ^K tetramer is quite limited ($\sim 488\text{\AA}^2$ for interface 1, $\sim 510\text{\AA}^2$ for interface 2), which likely makes for fragile interactions between SpoIVFB:Pro- σ^K monomers that are largely mediated by lipid/detergent."

9. The authors performed two reconstructions of WT and E44Q SpoIVFB: Pro- σ^K complexes and compared the final cryo-EM structures of WT and E44Q SpoIVFB:Pro- σ^K complexes and claimed that E44Q SpoIVFB:Pro- σ^K is approximately one-half the size of the tetrameric WT complex. However, there are no proper figures where the reader can see the clear difference between WT and E44Q SpoIVFB:Pro- σ^K complexes. A proper figure should be included in the manuscript.

Main Figure 2 shows the final cryo-EM map of WT (Fig. 2A) and E44Q (Fig. 2C) SpoIVFB:Pro- σ^K complexes. As can be seen from this figure the E44Q reconstruction clearly contains one half of the protein components as the WT reconstruction.

10. The authors focused on different oligomeric states of WT and E44Q SpoIVFB:Pro- σ^K in the entire manuscript. However, no proper biophysical experiments were performed to validate their claim. It will be better to perform SEC-MALS or other biophysical experiments to support their claim. It will also be better to perform 2D class averages of dimeric or trimeric 3D structures and support their claim.

Analysis of oligomeric states through methods such as SEC-MALS is difficult as the protein-detergent complexes for dimers, trimers, and tetramers elute within a similar volume (single peak) through size-exclusion chromatography (Suppl. Fig. S1B-E). Furthermore, proper SEC-MALS analysis of protein-detergent complexes requires an additional refractive index detector to separate out contributions from the detergent versus protein for overall mass calculations. Such analysis is non-trivial and requires specialized gel-filtration

equipment equipped with all three detectors (UV, MALS, RI), and complicated data processing to deconvolute the signal for different oligomeric species eluting in essentially the same elution volume peak. On the contrary, the cryo-EM experiment provides direct visualization of the different oligomeric species present in the sample, as is demonstrated in the 2D averages provided in Supplemental Fig. S2B. 2D averages for the different species are shown in main Figure 2B&D, Suppl. Fig S2B, and Suppl. Fig S5B.

11. More severe problem, the authors didn't mention how one single mutation, E44Q, could abolish the formation of trimers and tetramers.

We do not believe that the E44Q mutation has any effect on the oligomeric state of SpoIVFB. Rather, the differences in oligomeric state observed between the two datasets is likely a consequence of fragile interfaces between monomer subunits (much of which is mediated by detergent/lipid), which leads to prep-to-prep variability in the proportions of oligomeric states in the final purified sample.

12. Basically, the non-membrane part of the SpoIVFB:Pro- σ^K complex looks higher resolution than the TM region. Authors should perform local resolution calculations. RMSD $\sim 3.5\text{\AA}$ is a bit higher side. Several helix and loop regions of the non-membrane part of the SpoIVFB:Pro- σ^K complex of both maps are not fitted properly. Even at the high threshold, the helix and loop regions of the atomic model are outside of the EM density map (see the attachment). I am concerned about the model building. When EM densities are not visible, how accurate is an atomic model? Also, at higher thresholds, many extra densities are observed. In these circumstances, how confident were the authors about LMNG densities? The author should show the map-to-model fitting, angular distribution, and FSC with and without a mask.

The TM region of the map is higher resolution than the soluble region as reflected in the local resolution maps that were supplied in Suppl. Fig. S3C and S6C. In addition, angular distributions and FSC between half-maps with and without a mask were provided in Suppl. Fig. S3A and S6A, and the model-vs-map FSC at a proper resolution cutoff of 0.5 was reported in Suppl. Table S1 demonstrating concordance with the half-map FSC calculated at the gold-standard cutoff of 0.143. We are quite confident in the model building and fit to the cryo-EM map as exemplified for different regions of the model in Suppl. Fig. S3D-F and S6D-F, as well as the fit of LMNG to the experimental map in Fig. 4A&B.

The regions of the map/model that the reviewer points to in the attachment correspond to the lower resolution regions (C-terminus of Pro- σ^K , and loops between TM helices) as indicated by the local resolution calculation in Suppl. Fig. S3C. As the resolution typically varies across cryo-EM maps, one single post-processing low-pass filter resolution and B-factor may not be suitable for viewing all regions of the map. For these reasons the half-maps from final map refinement were deposited alongside the combined and sharpened map. With half-maps available, readers of the manuscript and viewers of the maps/models can apply various filter resolutions, B-factors, and other postprocessing methods of their choosing to view the map at different thresholds. The figure below shows the unfiltered and non-sharpened map with clear features of the regions that may be more difficult to view in maps with a higher B-factor.

13. The authors claimed that H43, H47, and D137 of SpoIVFB are conserved with those of mJ2P; the cryo-EM map of WT SpoIVFB:Pro-σ^K does not show any appreciable density for an ion being coordinated by these residues (Fig. 3B). However, in figure 3B represented Zn is binding with H43, H47, and D137. This is very misleading.

We apologize if the reviewer found this figure to be misleading, but we were very careful to explain the lack of zinc density in the manuscript text. Figure 3B was designed to demonstrate that indeed there is no appreciable density for zinc in the SpoIVFB:Pro-σ^K map. We explicitly state in the figure legend “The cryo-EM map demonstrates that no zinc ion was co-purified with WT SpoIVFB:Pro-σ^K.” For these reasons, we respectfully disagree that the figure is misleading, as it clearly demonstrates that the zinc present in the mJ2P structure is absent from the SpoIVFB:Pro-σ^K cryo-EM maps.

14. The authors also claimed that they were able to co-purify WT SpoIVFB with full-length unprocessed Pro-σ^K despite displaying robust proteolytic activity in *E. coli*. Also, extra Zn ions do not activate the proteolytic activity. How does this complex, if physiologically relevant? The author should perform an enzyme activity assay to demonstrate that the entire complex is enzymatically active.

We have made extensive efforts throughout the manuscript (see third to last paragraph in Discussion starting on line 379 to elaborate upon the situation regarding the lack of zinc ion (and thus proteolytic activity) in purified preparations of SpoIVFB:Pro-σ^K. Please also see the response to point #1 above for further clarification on the relevance of the structures presented here.

15. Based on the previous report and their experiment, the mentioned “H206A and H209A variants processed Pro-σ^K comparable to WT SpoIVFB”. However, it is unclear what they wanted to conclude from this experiment.

A previous report (Halder, S., Parrell, D., Whitten, D., Feig, M. & Kroos, L. Interaction of intramembrane metalloprotease SpoIVFB with substrate Pro-σ^K. PNAS. 2017) investigated the role of H206 and F209 in mediating Pro-σ^K cleavage and found impaired cleavage with the Ala variants. However, this previous investigation involved co-expressing the SpoIVFB variants and Pro-σ^K from independent plasmids. In contrast, in the current manuscript both proteins were encoded in a single plasmid under control of separate T7 promoters. In the single plasmid system we consistently observed more robust Pro-σ^K cleavage in the *E. coli*

based cleavage assay, including for the H206A and F209A variants that showed processing levels comparable to WT protein (in contrast to what was observed in the dual plasmid system of *Halder. et al. 2017*).

16. The cryo-EM data and MD simulation data showed that the membrane plays a role in stabilizing the complex. What type of lipid can play a role in enzyme stabilization? The authors can elaborate on that. Additionally, authors' biochemical data showed protein is inactive in the presence of detergent. How do authors justify lipid can play a role in enzyme stabilization and activation.

Please see response to reviewer #3 questions #1-3 above. We have performed additional simulations in a modified membrane composition, and expanded our results/discussion to include further elaboration upon lipid observed near the membrane reentrant loop and the pro-sequence of Pro- σ^K . Both the cryo-EM data and molecular simulations support that lipids may occupy the space above the membrane reentrant loop, suggesting that such lipids may be important in the overall substrate capture and cleavage process. More detailed biochemical work could certainly provide additional insight into such a potential role of lipids. However, such analysis will require development of a robust *in-vitro* cleavage assay, which has thus far not been possible with the material that we purify for cryo-EM analysis (due to lack of zinc). For this reason, we simply highlight the potential role that lipids may play in stabilizing the SpoIVFB: Pro- σ^K interaction as gleaned from the cryo-EM structures and molecular simulations. We have softened the statement (line 320) to say...

“Both the cryo-EM structure and MD simulations seem to support a potential role of membrane lipids in helping to stabilize the interaction and conformation of Pro- σ^K bound to SpoIVFB, possibly facilitating the overall proteolytic cleavage mechanism.”

REVIEWER COMMENTS

Reviewer #1 (Remarks to the Author):

The authors have addressed all of my concerns. And addressed most of the concerns of the other three reviewers. I think the manuscript is ready for publication and fully support its acceptance.

Reviewer #2 (Remarks to the Author):

The authors have thoroughly addressed all of my concerns. This is one of the most exciting studies I have read so far this year.

Reviewer #3 (Remarks to the Author):

The author responded to all my points with great detail and made a significant effort to implement all my comments. The manuscript was already very well organized, and the additional revisions have strengthened it even further. I also appreciated the submission of the simulations on Zenodo. This is the direction the field should take, and the authors are contributing to its evolution. I would recommend publishing it as is.

Reviewer #4 (Remarks to the Author):

The revised manuscript entitled "Substrate Engagement by the Intramembrane 1 Metalloprotease SpoIVFB" by Orlando et al. has modified the manuscript. However, I'm afraid I have to disagree with some of the answers. For questions 9 and 10, I understand it will be difficult for the authors to address the oligomeric states using biophysical techniques. At least the authors can perform biochemical experiments to show E44Q SpoIVFB: Pro- σ K complexes size is half of Wt and formed dimers, trimers, and tetramers. From cryo-EM 2D averages and 3D classification, it is difficult to confirm the protein is dimers, trimers, and tetramers when SEC can't segregate different populations (monomers, dimers, trimers, and tetramers) properly. Most of the time, researchers performed biochemical/biophysical experiments to validate different oligomeric states of the protein. So, my recommendation is that authors should perform some biochemical experiments to validate the mixed population of SpoIVFB:Pro- σ K. I am not sure how good it is to comment that they can measure protein content (E44Q reconstruction clearly contains one-half of the protein components as the WT reconstruction) and molecular weight from the 2D and 3D structures.

The authors mentioned in lines 103-105 that "Despite extensive efforts, we were unable to reconstitute SpoIVFB enzymatic activity and Pro- σ K cleavage in vitro using these detergent-solubilized preparations of WT SpoIVFB and Pro- σ K. An E44Q mutation in the HEXXH motif of SpoIVFB completely abolishes the proteolytic activity of the enzyme 6, 7." Very confused with this comment. Does their experimental data show this or previously reported?

Again, lines 142-148, in contrast to what was observed with WT SpoIVFB:Pro- σ K complexes, initial 2D and 3D classification of E44Q SpoIVFB:Pro- σ K particles showed a clear distribution of dimers and a lack of higher oligomeric species (Suppl. Fig. S5A-C). By comparing the final cryo-EM reconstructions of WT and E44Q SpoIVFB:Pro- σ K complexes, it becomes readily apparent that E44Q SpoIVFB:Pro- σ K is approximately one-half the size of the tetrameric WT complex (Fig. 2B)..... Based on this comment, this suggests that E44Q mutation at SpoIVFB:Pro- σ K contributes to dimer; therefore, how does author justify their answer of Q11 "We do not believe that the E44Q mutation has any effect on the oligomeric state of SpoIVFB."

Q12, The authors replied, "With half-maps available, readers of the manuscript and viewers of the maps/models can apply various filter resolutions, B-factors, and other postprocessing methods of their choosing to view the map at different thresholds. The figure below shows the unfiltered and non-sharpened map with clear features of the regions that may be more difficult to view in maps with a higher B-factor." It is a bit difficult for every reader to sharpen the map and fit the model before they read the paper. The reviewer also considers the map (b-factor sharpened map) that the authors suggested or deposited in EMDB. In rebuttal, the authors presented two images at a higher threshold to validate their fitting is good. Please see my attachment, which contains two new figures. The figures were prepared as the authors mentioned. At the filtered map (exactly represented as authors mentioned in the rebuttal figure of Q12), at the high threshold, the helix is engulfed by the cryo-EM map; however, a considerable amount of extra densities at the center of the cryo-EM map (fig 1a, blue circle); authors were unable to dock or build any atomic model in this extra densities which appeared due to high threshold values. Also, building a model from this filtered map is extremely difficult due to low-resolution features (fig 1a, red circle). Now, the same orientation sharpens map (fig 2) was fitted with the atomic model, which was built by authors based on their cryo-EM map; the authors claimed they are very confident about model building. Then why do several areas not fit in the EM density map? At least the atomic model built from the cryo-EM map should fit properly into the cryo-EM map. It will be better to build the model carefully.

Additionally, the authors mentioned that the differences in oligomeric state observed between the two datasets are likely a consequence of fragile interfaces between monomer subunits (much of which is mediated by detergent/lipid), which leads to prep-to-prep variability in the proportions of oligomeric states in the final purified sample. If there is a prep-to-prep to variability, how reproducible are the oligomeric states of SpoIVFB and its structure? Also, the authors commented, "fragile interfaces between monomer subunits." How fragile are they? Did they measure the interface areas? FSC curve of the final 3D reconstructions of tetrameric WT SpoIVFB:Pro- σ K and dimeric E44Q SpoIVFB:Pro- σ K look very odd, especially 3D reconstructions of tetrameric WT SpoIVFB:Pro- σ K. The authors may argue that due to detergent, there is a dip due to disordered detergent. However, FSC values are below 0.3 FSC, which is slightly odd. It may be possible that there is a substantial duplicate particle that represents unhealthy FSC. I will recommend cross-checking and deleting the duplicate particles. It would also be better to show the 3D-FSC. Additionally, there are some missing regions in angular distribution plots of particles in the final reconstruction of WT SpoIVFB:Pro- σ K. Is there any preferred orientation?

I have asked the author to show the map-to-model fitting. However, the authors did not show map-to-model FSC. However, FSC curves between the model and the cryo-EM map should be presented here.

Fig 1

Low resolution,
at high threshold, this area
is not suitable for model building.

At higher threshold, extra densities are visible. Thus, this threshold is not suitable for model fitting. **Which threshold do we use to fit the model?** We should use an extremely high threshold to fit atomic model. If we do so, we should explain the extra densities as well

Fig 2

Same map at same orientation but b-factor sharpened_map. We have to use higher threshold to fit the helix (orange ellipse). However, even that high threshold, several densities of atomic model are out side (cyan Helix and cyan ellipse)...

How atomic model built from cryo-em map can't fit properly (cyan ellipse region). It should be identical after model building if model building is accurate.

Also, at higher threshold, red circle region is unassigned densities. We should use a threshold

Where these extra densities are disappeared.

Dear *Nature Communications* Editorial Staff,

Thank you for facilitating the review of our manuscript titled "Substrate Engagement by the Intramembrane Metalloprotease SpoIVFB". We are very pleased that after initial revision three reviewers are fully supportive of publication. We have carefully analyzed the comments below from reviewer number four, and have added additional validation metrics of our cryo-EM maps/models, and also edited the atomic models to address problematic areas of the model/map fit highlighted by the reviewer. Below are point-by-point responses to the critiques raised by reviewer #4. The original comments and suggestions from the reviewers are in *italicized and indented text*. Our responses and any corrective actions that have been taken with the manuscript are in **bold beneath each reviewer critique**.

Reviewer #1 (Remarks to the Author):

The authors have addressed all of my concerns. And addressed most of the concerns of the other three reviewers. I think the manuscript is ready for publication and fully support its acceptance.

Reviewer #2 (Remarks to the Author):

The authors have thoroughly addressed all of my concerns. This is one of the most exciting studies I have read so far this year.

Reviewer #3 (Remarks to the Author):

The author responded to all my points with great detail and made a significant effort to implement all my comments. The manuscript was already very well organized, and the additional revisions have strengthened it even further. I also appreciated the submission of the simulations on Zenodo. This is the direction the field should take, and the authors are contributing to its evolution. I would recommend publishing it as is.

We would like to thank reviewers 1-3 for their time and expertise in reviewing our manuscript, which we believe has been significantly strengthened through their suggestions.

Reviewer #4 (Remarks to the Author):

The revised manuscript entitled "Substrate Engagement by the Intramembrane 1 Metalloprotease SpoIVFB" by Orlando et al. has modified the manuscript. However, I'm afraid I have to disagree with some of the answers. For questions 9 and 10, I understand it will be difficult for the authors to address the oligomeric states using biophysical techniques. At least the authors can perform biochemical experiments to show E44Q SpoIVFB: Pro- σ K complexes size is half of Wt and formed dimers, trimers, and tetramers. From cryo-EM 2D averages and 3D classification, it is difficult to confirm the protein is dimers, trimers, and tetramers when SEC can't segregate different populations (monomers, dimers, trimmers, and tetramers) properly. Most of the time, researchers performed biochemical/biophysical experiments to validate different oligomeric states of the protein. So, my recommendation is that authors should perform some biochemical experiments to validate the mixed population of SpoIVFB:Pro- σ K. I am not sure how good it is to comment that they can measure protein content (E44Q reconstruction clearly

contains one-half of the protein components as the WT reconstruction) and molecular weight from the 2D and 3D structures.

We agree that biophysical/biochemical experiments should be performed to validate oligomeric states of protein complexes when such quaternary arrangements are important for biological function. However, in the case of SpoIVFB:pro- σ^K complexes presented here we do not believe that the oligomers observed are biologically relevant. To be as transparent as possible on this matter we stated in the manuscript...(line numbers refer to the revised manuscript with track changes)

Line 130: “we believe that the oligomeric arrangement of SpoIVFB:Pro- σ^K monomers in the tetramer is likely an artifact of detergent solubilization, rather than a representation of an oligomeric state that is sampled in vivo.”

Line 141: “difference in oligomeric state, which we believe arises simply from variations between protein preparations and fragile monomer interfaces”

Line 396: “We believe that these oligomeric states are likely an artifact of detergent solubilization and weak lipid/detergent interactions between monomers in the purified protein preparations”

Line 401: “Despite these previous reports of SpoIVFB oligomerization, it seems unlikely that the oligomeric states we observe here with cryo-EM represent functional oligomeric states that are sampled in sporulating *B. subtilis*.”

In our own hands, prep-to-prep variability in oligomeric states is apparent (see below), but we do not believe the oligomers observed are biologically relevant. More importantly, the oligomeric states presented here are incompatible with models of the inhibited complex with BofA and SpoIVFA (presented in previous response to reviewers). All lines of evidence suggest that the oligomeric states observed here are artifacts of membrane extraction and purification in detergent. For this reason, we feel that validating oligomeric states through methods other than size-exclusion chromatography and single particle cryo-EM presented herein is unlikely to provide further insights of biological relevance.

The authors mentioned in lines 103-105 that "Despite extensive efforts, we were unable to reconstitute SpoIVFB enzymatic activity and Pro- σ^K cleavage in vitro using these detergent-solubilized preparations of WT SpoIVFB and Pro- σ^K . An E44Q mutation in the HEXXH motif of SpoIVFB completely abolishes the proteolytic activity of the enzyme 6, 7." Very confused with this comment. Does their experimental data show this or previously reported?

Data showing the lack of protease activity in purified SpoIVFB:Pro- σ^K was not shown in the original manuscript. We have now included Supplemental Figure 1F to show the lack of *in vitro* activity both in the presence and absence of added exogenous zinc. Data demonstrating that the E44Q mutation abolishes protease activity is shown in multiple references throughout the manuscript, as well as in Figure 4C, 5B, S1D, and S7D.

Again, lines 142-148, in contrast to what was observed with WT SpoIVFB:Pro- σ^K complexes, initial 2D and 3D classification of E44Q SpoIVFB:Pro- σ^K particles showed a clear distribution of dimers and a lack of higher oligomeric species (Suppl. Fig. S5A-C). By comparing the final cryo-EM reconstructions of WT and E44Q SpoIVFB:Pro- σ^K complexes, it becomes readily apparent that E44Q SpoIVFB:Pro- σ^K is approximately one-half the size of the tetrameric WT complex (Fig. 2B)..... Based on this comment, this suggests that E44Q mutation at SpoIVFB:Pro- σ^K

contributes to dimer; therefore, how does author justify their answer of Q11 "We do not believe that the E44Q mutation has any effect on the oligomeric state of SpoIVFB."

We believe that the mutation of E44Q which is buried within the core of the enzyme and far from any oligomeric interfaces does not cause long-range structural perturbations that would alter the relative oligomeric state. Rather, we believe the difference in oligomeric state between WT and E44Q preparations simply arises from prep-to-prep variability of detergent solubilized complexes which is caused by fragile monomer interfaces that are mediated by lipid/detergent and subject to dissolution during membrane solubilization and detergent washes during purification. In the line immediately after the one pointed out by the reviewer above we state...

Line 141: "Aside from the difference in oligomeric state (between WT and E44Q), which we believe arises simply from variations between protein preparations and fragile monomer interfaces,"

Q12, The authors replied, "With half-maps available, readers of the manuscript and viewers of the maps/models can apply various filter resolutions, B-factors, and other postprocessing methods of their choosing to view the map at different thresholds. The figure below shows the unfiltered and non-sharpened map with clear features of the regions that may be more difficult to view in maps with a higher B-factor." It is a bit difficult for every reader to sharpen the map and fit the model before they read the paper. The reviewer also considers the map (b-factor sharpened map) that the authors suggested or deposited in EMDB. In rebuttal, the authors presented two images at a higher threshold to validate their fitting is good. Please see my attachment, which contains two new figures. The figures were prepared as the authors mentioned. At the filtered map (exactly represented as authors mentioned in the rebuttal figure of Q12), at the high threshold, the helix is engulfed by the cryo-EM map; however, a considerable amount of extra densities at the center of the cryo-EM map (fig 1a, blue circle); authors were unable to dock or build any atomic model in this extra densities which appeared due to high threshold values. Also, building a model from this filtered map is extremely difficult due to low-resolution features (fig 1a, red circle). Now, the same orientation sharpens map (fig 2) was fitted with the atomic model, which was built by authors based on their cryo-EM map; the authors claimed they are very confident about model building. Then why do several areas not fit in the EM density map? At least the atomic model built from the cryo-EM map should fit properly into the cryo-EM map. It will be better to build the model carefully.

The problematic regions that the reviewer is pointing to are the C-terminus of Pro- σ^K (residues ~119-126) and the loop region in the SpoIVFB CBS domain corresponding to residues ~250-270. Both of these regions are at the extreme periphery of the complex in highly solvent exposed areas, which likely leads to greater flexibility and diminished resolutions in these regions. This diminished resolution is apparent in the local resolution plots in Suppl. Fig. S3D & S6D and the figure shown below. In order to clarify that these regions are of significantly lower resolution such that reliable sidechain placement is difficult or impossible we have truncated the sidechains of residues at the C-terminus of Pro- σ^K (residues 119-126) and the loop in the SpoIVFB CBS domain (residues 253-263) to the C $_{\beta}$ carbon (ie: polyalanine). The figures below demonstrate the map/model in these regions before and after sidechain truncation. The truncated and re-refined models for WT and E44Q datasets have been uploaded to the PDB, and statistics in Supplemental Table S1 have been updated to reflect changes from these re-refined models.

We have added the following lines to the manuscript to clarify that these regions of the maps are of lower quality, and model building included truncation of sidechains to the C $_{\beta}$ carbon in these regions...

Line 484: “In both the WT and E44Q maps of SpoIVFB:Pro- σ^K the regions encompassing the C-terminus of Pro- σ^K (residues 119-126) and a loop in the SpoIVFB CBS domain (residues 253-263) were of particularly lower resolution than the rest of the complex (Suppl. Fig S3D and S6D). Sidechains of residues in these regions were trimmed back to the C $_{\beta}$ carbon.”

Additionally, the authors mentioned that the differences in oligomeric state observed between the two datasets are likely a consequence of fragile interfaces between monomer subunits (much of which is mediated by detergent/lipid), which leads to prep-to-prep variability in the proportions of oligomeric states in the final purified sample. If there is a prep-to-prep to variability, how reproducible are the oligomeric states of SpoIVFB and its structure? Also, the authors commented, "fragile interfaces between monomer subunits." How fragile are they? Did they measure the interface areas?

In our own experiments leading up to final dataset collection we have observed variations in the relative ratios of different oligomeric states. Below is an example of an early dataset of WT SpoIVFB:pro- σ^K that showed a predominant dimer species that could be reconstructed to sub-nanometer resolution. This early screening dataset was collected on a Talos Arctica with a Falcon III detector and reached significantly lower resolution than the datasets presented in the manuscript.

Nevertheless, this data clearly demonstrates the prep-to-prep variability in oligomeric state, as even a dimeric species of WT SpoIVFB:pro- σ^K can be reconstructed similar to that presented for E44Q in the manuscript. In the manuscript we measured the interaction surfaces between protomers in the complex and stated...

Line 127: "In total, the physical interaction between monomers in the WT SpoIVFB:Pro- σ^K tetramer is quite limited ($\sim 488\text{\AA}^2$ for interface 1, $\sim 510\text{\AA}^2$ for interface 2)"

FSC curve of the final 3D reconstructions of tetrameric WT SpoIVFB:Pro- σ^K and dimeric E44Q SpoIVFB:Pro- σ^K look very odd, especially 3D reconstructions of tetrameric WT SpoIVFB:Pro- σ^K . The authors may argue that due to detergent, there is a dip due to disordered detergent. However, FSC values are below 0.3 FSC, which is slightly odd. It may be possible that there is a substantial duplicate particle that represents unhealthy FSC. I will recommend cross-checking and deleting the duplicate particles. It would also be better to show the 3D-FSC. Additionally, there are some missing regions in angular distribution plots of particles in the final reconstruction of WT SpoIVFB:Pro- σ^K . Is there any preferred orientation?

We have crosschecked and there are no duplicated particles. We have also now included the 3DFSC for both WT and E44Q in the supplemental material (Supplemental Figure S3C and S6C) showing the sphericity (0.963 for WT, and 0.866 for E44Q) indicating a mostly isotropic

reconstruction. As indicated in Supplemental Figure S3B and S6B there is slight preferred orientation of particles in the dataset, but this does not manifest as significant visible streaking or other directional anomalies in the final reconstructed maps.

I have asked the author to show the map-to-model fitting. However, the authors did not show map-to-model FSC. However, FSC curves between the model and the cryo-EM map should be presented here.

Map-vs-model FSC curves have been added to Supplemental Figure S3A and S6A. The map-vs-model FSC at a cutoff value of 0.5 is also reported in the Supplemental Table 1.